# A comparative study of plant water extraction methods for isotopic analyses: Scholander-type pressure chamber vs. cryogenic vacuum distillation

Giulia Zuecco[1], Anam Amin[1], Jay Frentress[2], Michael Engel[2], Chiara Marchina[1], Tommaso Anfodillo[1], Marco Borga[1], Vinicio Carraro[1], Francesca Scandellari[3], Massimo Tagliavini[2], Damiano Zanotelli[2], Francesco Comiti[2], and Daniele Penna[4]

[1]Department of Land, Environment, Agriculture and Forestry, University of Padova, Italy
[2]Faculty of Science and Technology, Free University of Bozen-Bolzano, Italy
[3]Isotracer s.r.l., Bologna, Italy
[4]Department of Agriculture, Food, Environment and Forestry, University of Florence, Italy
**Correspondence:** Giulia Zuecco (giulia.zuecco@unipd.it)

**Abstract.**

Recent tracer-based studies using stable isotopes of hydrogen and oxygen showed that different methods for extracting water from plant tissues can return different isotopic composition due to the presence of organic compounds, and the extraction of different plant water domains. One of the most used methods to extract plant water is the cryogenic vacuum distillation (CVD), which tends to extract total plant water. Conversely, the Scholander-type pressure chamber (SPC), which is commonly used by tree physiologists to measure water potential in plant tissues and determine plant water stress, likely accesses only the more mobile plant water (i.e., xylem and inter-cellular water). However, only few studies reported the application of SPC to extract plant water for isotopic analyses, and therefore, inter-method comparisons between SPC and CVD are urgently needed.

In this work, we analyzed the variability in the isotopic composition of plant water extracted by SPC and CVD, also considering the potential variability in the isotopic signature of the plant water extracted from various tissues by CVD (i.e., leaves, twig without bark, twig with bark, twig close to the trunk of the tree, and wood core), and from different plant species (i.e., alder, apple, chestnut and beech). The extraction of plant water by SPC is simple, can be carried out in the field, and it does not require specific laboratory work as in case of CVD. However, the main limitation of SPC is the very small water volume that can be extracted from the lignified twigs during conditions of water deficit, compared to CVD.

Our results indicated that plant water extracted by SPC and CVD were significantly different. The difference in the isotopic composition obtained by the two extraction methods was smaller in the beech samples compared to alder, apple and chestnut samples. The isotopic signature of alder, apple and chestnut plant water extracted by SPC was more enriched in $\delta^2$H and $\delta^{18}$O, respectively, than the samples obtained by CVD. We conclude that plant water extraction by SPC is not an alternative for CVD, as SPC likely extracts mostly the mobile plant water, whereas CVD tends to retrieve all water stored in the sampled tissue, from both living and dead cells. However, studies aiming to quantify the relative contribution of the soil water sources to transpiration should rely more on the isotopic composition of xylem water (which is theoretically sampled by SPC), than

the isotopic composition of total plant water (sampled by CVD), which also contains a fraction of water that could be stored in plant tissues for a longer time.

*Keywords:* stable isotopes of hydrogen and oxygen; cryogenic vacuum distillation; Scholander-type pressure chamber; plant water; xylem water.

## 1   Introduction

Stable isotopes in the water molecule ($^2$H and $^{18}$O) have been extensively used as environmental tracers in hydrological studies to track water fluxes and estimate water flow pathways, mean residence times, and water storage (e.g., Dansgaard, 1953; Craig, 1961; Klaus and McDonnell, 2013). The development of low-cost and easy-to-use spectroscopic techniques for the collection and isotopic analysis of water samples at a high temporal resolution (e.g., Kerstel et al., 1999; Penna et al., 2010; von Freyberg et al., 2017) stimulated the application of stable isotopes to investigate water transfer in the soil-plant-atmosphere continuum (Brooks et al., 2010; McDonnell, 2014). An increasing number of studies has been recently conducted to better understand water dynamics, such as water uptake and evapotranspiration partitioning, in the soil-plant-low atmosphere continuum in different climates and in both natural (e.g., Allen et al., 2019; Dubbert et al., 2019; Liu et al., 2019a; Oerter et al., 2019; Qiu et al., 2019) and managed (agricultural and agroforest) (e.g., Aguzzoni et al., 2022; Liu et al., 2019b; Quade et al., 2019; Zhang et al., 2019; Penna et al., 2020) environments. Despite the rapid increase of the number of studies based on the stable isotope approach, only a small fraction of them focused on the comparison of two or more plant water extraction techniques (e.g., Millar et al., 2018; Fischer et al., 2019; Barbeta et al., 2022).

Ecohydrological studies relying on the isotopic signature of plant water require sampling methods that extract water representative of transpiration, and that do not alter the original isotopic composition of the plant material. This is still a critical aspect because a standardized and shared procedure and method for isotope-based ecohydrological studies has not been defined yet (Penna et al., 2018). Indeed, there is a variety of different techniques for the extraction of plant water, such as *in situ* direct vapour equilibration (Sprenger et al., 2015; Volkmann et al., 2016; Marshall et al., 2020), microwave extraction (Munksgaard et al., 2014), cryogenic vacuum distillation (Koeniger et al., 2011; Orlowski et al., 2013, 2016a), centrifugation (Peters and Yakir, 2008), and high-pressure mechanical squeezing (Böttcher et al., 1997). In addition, xylem water can be extracted by a syringe with a needle inserted in a pre-drilled hole (Zhao et al., 2016) or by a cavitron flow-rotor (Barbeta et al., 2022). Among these, cryogenic vacuum distillation (abbreviated in CVD thereinafter) is widely applied (Orlowski et al., 2018; Amin et al., 2020). During CVD, the soil or plant material is heated in a tube under a specified vacuum to evaporate the sample water, that afterwards is frozen and collected in a cryogenic trap (Koeniger et al., 2011; Orlowski et al., 2013). As such, this technique extracts the entire volume of water from plant tissues, including water within cell walls (Millar et al., 2018). This volume may include water that underwent fractionation processes and/or water with a different age, and stored in dead and living cells for days or weeks (Sprenger et al., 2019), so not only water that is transported at the time of the sampling. This might be a serious limitation in ecohydrological and physiological studies aiming at understanding water sources for plant transpiration as the isotopic composition of water stored in plant tissues for a long time is possibly different from that of xylem water. Moreover,

experimental evidence showed that different techniques might return different isotopic values due to intrinsic methodological differences (Beyer and Penna, 2021). CVD, in the case of soil water, was shown to reveal large differences in the isotopic composition of water extracted from soil samples by different laboratories, although strictly consistent procedures were applied (Orlowski et al., 2018). These authors also observed no clear trends in the results, and differences depended on the interplay of multiple factors, such as soil type and properties, soil water content, system setup, extraction efficiency, extraction system leaks, and each laboratory's internal accuracy.

Recently, Millar et al. (2018) performed a thorough comparison of six plant water extraction techniques (i.e., direct vapour equilibration, microwave extraction, two versions of CVD, centrifugation, and high-pressure mechanical squeezing) based on four plant portions of spring wheat (*Triticum aestivum*). The authors found marked differences among the measured isotopic compositions of plant water, with the CVD systems and the high-pressure mechanical squeezing producing waters more depleted in heavy isotopes compared to the other techniques. Particularly, Millar et al. (2018) associated the differences in the isotopic compositions of plant water to the ability of each extraction system to access different plant water domains. The authors argued that CVD, microwave extraction, centrifugation, and high-pressure mechanical squeezing could access all plant water domains (i.e., the more mobile xylem water and inter-cellular water, and the less mobile intra-cellular, cell wall, and organelle constrained water), whereas direct vapour equilibration could only extract the mobile xylem and inter-cellular water. Millar et al. (2018) concluded that, in terms of limited co-extraction of organic compounds and speed of sample throughput, the direct vapour equilibration outperformed CVD.

Fischer et al. (2019) proposed and described various low-tech plant water sampling and extraction techniques, and compared them to the CVD developed by Koeniger et al. (2011) for different plant species. Fischer et al. (2019) found that the new methods produced consistent and comparable results to those provided by CVD. However, these authors, due to the limited amount of plant material, could not assess the water domains accessed by the different methods for each plant type. Fischer et al. (2019) also showed that other factors, such as appropriate transport and storage of the samples from the field site to the laboratory, fast sample processing, and efficient workflows, significantly influenced the accuracy and precision of the measured isotopic composition.

Comparing different techniques for plant water extraction, understanding which water domain each method accesses, and whether isotopic fractionation occurs during the extraction process are becoming increasingly important, particularly when isotopic differences between plant water and the respective potential water sources used for transpiration are observed. Indeed, Barbeta et al. (2019) found that isotopic fractionation resulting in an unexpected depleted $\delta^2$H of xylem water complicated the identification and quantification of the water sources used by beech trees in a temperate forest in France. These authors recommended that future research should investigate the physico-chemical fractionation processes occurring in the unsaturated zone, and improve the understanding of the isotopic dynamics of water stored within the plant tissues. If plant water domains have distinct isotopic signatures, new techniques should be developed with the aim of extracting the target plant water domain. More recently, studies by Chen et al. (2020) and Barbeta et al. (2022) observed that $\delta^2$H of stem water extracted by CVD had a significant depletion compared to $\delta^2$H of xylem water (however, such large difference was not found for $\delta^{18}$O), but there was no significant $\delta^2$H offset between xylem and source waters. Barbeta et al. (2022) commented that factors leading to these $\delta^2$H

offsets are still unclear. Indeed, while Chen et al. (2020) proposed that the offset was due to the H exchange between organically bound deuterium in the wood material and liquid water during the CVD extraction, other studies (e.g., Zhao et al., 2016; Barbeta et al., 2020) suggested a possible effect of aquaporin- or surface-mediated within-stem water isotope heterogeneity on the $\delta^2$H offset. This $\delta^2$H bias in the CVD extraction can affect significantly the plant water source identification (Allen and Kirchner, 2022; Barbeta et al., 2022). The effect of the $\delta^2$H bias on the inference of the plant water sources can be substantial when the isotopic differences among the end members is small, whereas this effect is less marked when there is a large difference among the isotopic compositions of the end members (Allen and Kirchner, 2022).

Despite the number of studies that focused on the comparison of techniques for plant water extraction, previous and current research, so far has not considered ecophysiological-based methods that tree physiologists usually adopt to measure leaf water potential and determine plant water stress (e.g., Scholander, 1966; Meiri et al., 1975; Grossiord et al., 2017; Bowling et al., 2017). One of these methods, namely the Scholander-type pressure chamber (abbreviated in SPC thereinafter) is based on an external pressure to retrieve the mobile water transported within the xylem conduits to measure the plant water potential. Although SPC is widely used in plant water relations studies to measure plant water potential, it is not commonly applied to collect the extracted water for isotopic analyses. We have found only four published studies (Ellsworth and Williams, 2007; Penna et al., 2013; Gei$\beta$ler et al., 2019; Magh et al., 2020) that used SPC to extract plant water for isotopic analysis. One of them, Gei$\beta$ler et al. (2019), made a simple comparison between $\delta^{18}$O of water extracted by SPC and CVD in stem water samples collected from *Acacia mellifera* shrubs. Samples were 10 cm long and lignified, and, for SPC extraction, the authors removed leaves, bark, and green tissues to avoid contamination with phloem. Overall, Gei$\beta$ler et al. (2019) found no significant difference in the isotopic composition of the plant water extracted by the two methods. However, this analysis was based on the comparison of six samples and one plant species only, and more robust comparative tests are missing. Therefore, assessing potential differences in isotope data retrieved by using SPC and the CVD extraction techniques based on a larger number of samples and different plant species is urgently needed to find the best method to sample transpiration water.

In this study we assume that SPC extracts relatively mobile plant water only, in contrast to CVD, which accesses all plant water domains (Millar et al., 2018), and water potentially fractionated. Given that the relatively mobile plant water might have a different age and a different isotopic composition compared to the less mobile water domain (Sprenger et al., 2019), we hypothesize that SPC and CVD return significant differences in the isotopic composition of the extracted plant water. Therefore, our specific objectives are to: *i)* quantify the differences in the isotopic composition of plant water extracted by the two techniques, and *ii)* assess how differences in the isotopic composition are related to plant species or plant tissue type used for CVD.

## 2  Study sites and sampling

### 2.1  Ahr/Aurino

Samples from grey alder trees (*Alnus incana*) were taken at two sites in the riparian area of the Ahr/Aurino River in the Eastern Italian Alps (Fig. 1). The study site is located at about 882 m a.s.l. in the lower valley, where the typical valley form is U-shape.

The catchment is mostly composed of metamorphic (gneiss, micaschists) and magmatic (tonalite) rocks. The median diameter of sediment in the upper meter of soil in the former floodplain ranges from 0.3 to 0.5 mm (Andreoli et al., 2020). The climate is

125 cold temperate with an average annual air temperature of about 7.7 °C (period 1992-2018) and an average yearly precipitation amount of about 821 mm/yr (period 1972-2018). The Ahr/Aurino River regime is nivo-glacial (the glacierized area is about 4 %). Riparian vegetation mainly consists of mature (at the upstream site in Fig. 1) and juvenile patches (at the downstream site) of grey alder with a thick tall herb (*Rubus caesius*, *Glechoma hederacea* and *Urtica dioica*, *Sambucus nigra* shrubs and the vine *Humulus lupulus*). Gravel mining activities in the 1950s to the 1980s resulted in riverbed incision, and a floodplain being

disconnected from its channel (Campana et al., 2014).

The sampling campaign was carried out on 7 June 2017 during a period of prolonged water deficit. Due to logistic issues, and to collect samples when the transpiration fluxes were close to their minimum, plant water was collected after the sunset from four alder trees (two at the downstream and two at the upstream site) in the Ahr/Aurino study area. Samples for water extraction by SPC and CVD were taken from the same position in the trees.

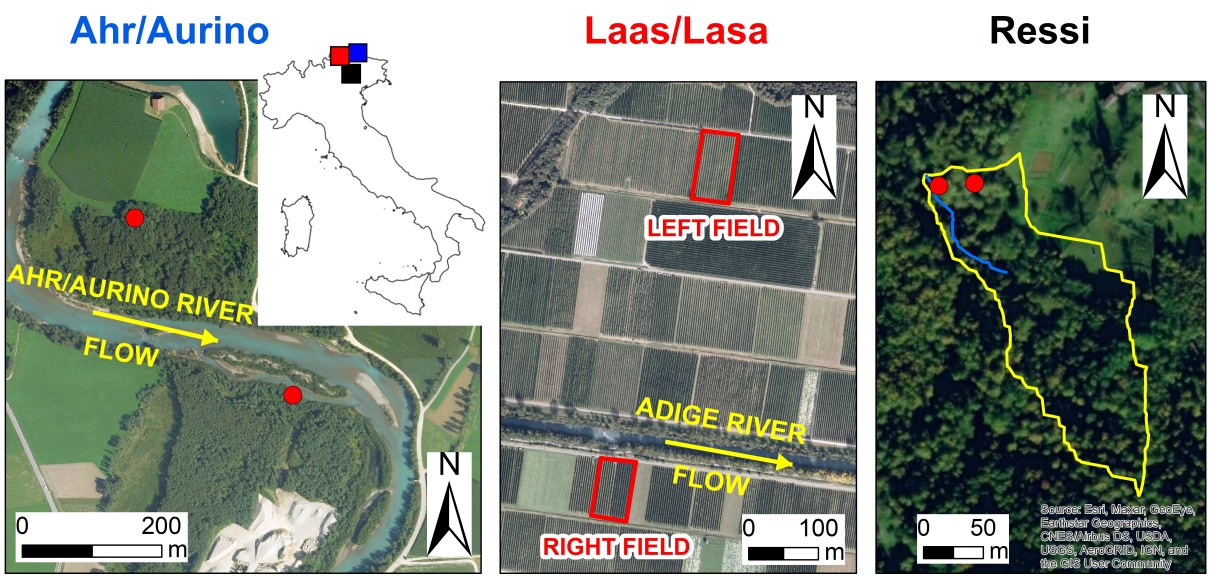

**Figure 1.** Aerial photos of the study sites, and location in northern Italy (blue: Ahr/Aurino; red: Laas/Lasa; black: Ressi). Red dots in Ahr/Aurino and Ressi indicate the approximate position of the sampled trees. In Ressi, the blue and the yellow solid lines mark the stream network and the catchment divide, respectively. Sources of aerial photos: ©Autonomous Province of Bolzano-South Tyrol (study sites: Ahr/Aurino and Lasa); Esri, "World Imagery" [basemap], scale not given, "World Imagery", August 14, 2020, http://www.arcgis.com/home/item.html?id=10df2279f9684e4a9f6a7f08febac2a9, (September 1, 2020) (study site: Ressi). The maps were made using Esri ArcGIS 10.7.1.

## 2.2 Laas/Lasa

Samples from cultivated apple trees (*Malus domestica*, cv. *"Pinova"* grafted on *"M9"* rootstock) were collected in two apple orchards in the Laas/Lasa (Vinschgau/Val Venosta region, South Tyrol; Fig. 1). The orchards are located at about 860 m a.s.l. on the right and left side of the river Etsch/Adige, with different distance from the river (50 m vs. 450 m, respectively). Within each orchard, a plot of about 400 m$^2$ was selected for sampling. The average annual precipitation recorded at the Laas/Lasa weather station (874 m a.s.l., operated by the Hydrographic Office of the Autonomous Province of Bozen-Bolzano) was approximately 480 mm (period 1989–2012) (Penna et al., 2021). Minimum average temperatures are below 0 °C during winter (from December to February), while maximum average temperatures can reach 24 °C in July (Penna et al., 2021). The soil in both orchards had a silty loam texture.

The sampling campaign was performed on 8 June 2017 during a period of water deficit. All samples were equally taken both at the right and the left field. Due to logistic issues and to collect samples when the transpiration fluxes were close to their minimum, samplings were carried out after the sunset. Samples for water extraction by SPC and CVD were taken from the same position in the trees.

## 2.3 Ressi

Samples from beech (*Fagus sylvatica*) and chestnut (*Castanea sativa*) trees were collected in the 2.4-ha Ressi catchment in the Italian pre-Alps (Fig. 1) (Zuecco et al., 2014, 2021). The catchment is located at the foothills of the eastern Italian Alps (elevation range: 598-721 m a.s.l.) and is densely vegetated. The climate is humid temperate and the average annual precipitation (period 1992-2007) recorded by a weather station approximately 4.5 km from Ressi is 1695 mm/yr (Penna et al., 2015). Monthly distribution of rainfall is bimodal with peaks in spring and fall. The mean annual temperature is 9.7 °C; on average the minimum monthly temperature is in January (1.2 °C) and the maximum in July (18.7 °C). The top 10 cm of the soil has a sandy clay loam texture; deeper in the profile the soil has a sandy clay texture (Penna et al., 2015; Zuecco et al., 2021).

The sampling campaign was carried out on 5 July 2017 during a period of prolonged water deficit. Samples for plant water extraction were retrieved at the sunrise from five beech and five chestnut trees at two sites in the lower part of the Ressi catchment. Samples for water extraction by SPC and CVD were taken from the same position in the trees. The sampling design aimed to replicate the sample collection in all the selected trees with the two investigated methodologies (Table 1). However, plant water extraction was not always possible by the SPC method because, in some cases (1 out of 5 chestnut samples, and 2 out of 5 beech samples), the extracted plant water volume was not always enough for isotopic analysis. In addition, we discarded some plant water samples extracted by CVD, affected by injection issues during the isotopic analysis, and for which we could not perform a second run of isotopic analyses due to the small water volume. Therefore, in this study we reported only the isotopic data relative to the plant water extracted by both methods (i.e., SPC and CVD) from the same trees (Table 1).

 # 3 Materials and methods

## 3.1 Extraction of plant water: the SPC method

The SPC is an instrument normally used by tree physiologists to measure water potential in plant tissues (e.g., Scholander, 1966; Meiri et al., 1975; Donovan et al., 2003; Grossiord et al., 2017). Typically, SPC is used to determine plant water potential (Cochard et al., 2001) and, being a proxy of the tissue water content, it can signal the occurrence of water deficit. The basic working principle is the use of an external pressure to retrieve the water within the xylem conduits (Scholander et al., 1965; Turner, 1981; Castro Neto et al., 2004) (Fig. 2). In this study, we used the SPC to force water out of the twigs, and collect water samples for isotopic analyses. The sampling material consisted of lignified twigs, with a diameter ranging between 3 and 6 mm. In agreement with Penna et al. (2013, 2021), we kept the bark and the leaves attached to the twig.

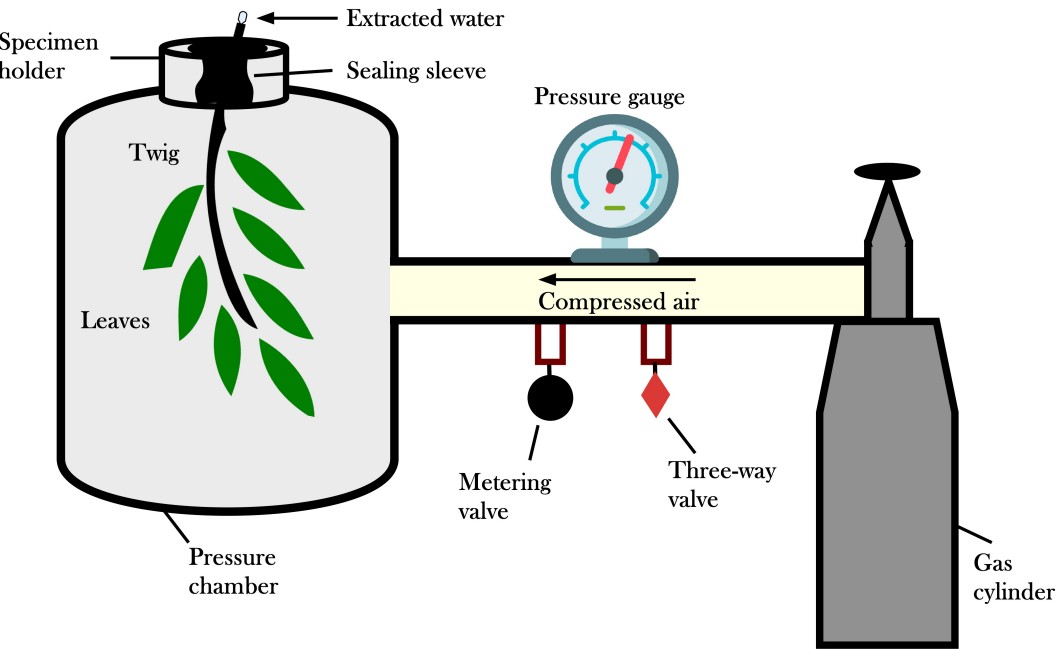

**Figure 2.** Sketch showing the technical setup of the Scholander-type pressure chamber (SPC).

The set up for the plant water extraction consisted of a lignified twig with one or more leaves sealed inside the chamber, while the cut end of the twig was exposed to the atmosphere (Fig. 2). After connecting the SPC to the gas tank, a three-way control valve was turned to pressure and the metering valve was slowly opened to begin to pressurize the unit. A pressure equal to the water potential was applied until water sprang from the cut end of the twig. For the SPC plant water extractions, we used a Pump-Up Chamber with a 2.0 MPa gauge (PMS Instrument Company, Oregon, USA) in Ahr/Aurino and Laas/Lasa, and a

SAPS II portable plant water status console (model 3115) with a 4.0 MPa gauge (Soilmoisture Equipment Corp., California, USA) in Ressi. The plant water was collected in 2 ml glass vials (which were immediately capped) by using pipettes or with the help of gravity (SPC was put on its side).

To extract water from the plant tissues, we had to apply a pressure of about 0.5 MPa in Laas/Lasa, 1.0-1.5 MPa in Ahr/Aurino, and 3.0 MPa in Ressi; the different plant water potentials indicate that the sampled vegetation in Ressi suffered higher water deficit conditions than the sampled plants in Ahr/Aurino and Laas/Lasa. The lower water deficit in Laas/Lasa than in the other two sites can be explained by the irrigation of the apple orchards during dry periods. The water extraction by the SPC method ended when we collected all the water flowing out from the twigs. The duration of the extraction was different among the samples (due to the different water deficit conditions), but we kept it as short as possible (less than 10 min) to minimize the evaporation. Note that the sampled volume was smaller than 200 $\mu$l during the sampling campaigns carried out for this study (Table 1). All the samples were stored in a fridge at 4 °C until the isotopic analyses.

## 3.2    Extraction of plant water: the CVD method

To extract plant water by CVD, we collected samples from different plant tissues, along a branch, in 12 ml glass Exetainer® vials (Labco Ltd., UK). After cutting the twigs from the trees, we removed all the leaves and other green tissues close to the leaves. Some of these leaves were collected in the vials for the extraction by CVD (i.e., CVD-L samples). CVD-L samples were used to determine the isotopic composition of the leaf water. The twig samples were lignified and approximately 85 mm long and 3-6 mm thick. For some of the twig samples, we kept the bark (i.e., twig with bark samples, abbreviated in CVD-TwB), whereas for others, bark was peeled using a knife (i.e., twig without bark samples, abbreviated in CVD-T; Table 1). CVD-extracted plant water (i.e., CVD-TwB) and plant water deprived of the phloem tissue (i.e., CVD-T) were used to assess the differences in the isotopic composition with the plant water extracted by the SPC method. In Ahr/Aurino and Laas/Lasa study sites, the diameters at the breast height of the alder and apple trees allowed for the collection of wood core samples (abbreviated in CVD-WC), that were retrieved by an increment borer (phloem tissue was removed, and the heartwood was not collected during the samplings). In Ressi, instead of wood cores (the sampling was not possible because of the small tree diameters and the location in a private land), we collected additional twig samples that were located close to the trunk (abbreviated in CVD-TcT). For these samples, we removed the bark by a knife. CVD-TcT samples were supposed to have older tissues and more dead cells than the twigs collected closer to the leaves (i.e., CVD-T and CVD-TwB).

The plant water volume of CVD samples was larger than volume of SPC samples (Table 1), with a minimum of 100 $\mu$l (three CVD-WC samples from alder trees) and a maximum of 2690 $\mu$l (a CVD-L sample from an apple tree). All the samples for CVD were stored in a fridge at 4 °C until the extraction and the consequent isotopic analyses.

The CVD was performed in the laboratory of the Faculty of Science and Technology of the Free University of Bozen-Bolzano (Italy) (Fig. 3). The CVD system was developed based on the method of Koeniger et al. (2011). The system consists of independent extraction-collection units, where the capped sample vial was connected to a second empty vial (hereafter collection vial) using a 1.56 mm diameter stainless steel capillary tube. After the preparation of the extraction-collection unit, the samples were frozen by immersing the sample vials in liquid nitrogen (approximately at –196 °C) to prevent loss of water

vapour during evacuation (vials were evacuated to a pressure of 0.95 kPa). The sample vials were then loaded in an aluminum block (with slots for 10 vials) and heated to a temperature of 200 °C (Fig. 3). At the same time, during the extraction process,

the bottom of the collection vials was immersed into the liquid nitrogen trap, which allowed for the evacuation of the sample from the heated vial and its condensation in the collection vial. All the individual plant samples were extracted at a temperature of 200 °C for an extraction time of 15 min per sample. A heat gun (at 300 °C) was used at the end of each extraction round to remove from the steel tube any water vapour trapped in the capillary tube. After the water had been quantitatively transferred from the plant tissue to the collection vial, vials were removed from the liquid nitrogen cold trap, defrosted at room temperature

under perfect sealed conditions and stored in a refrigerator after labelling, and tightly wrapped with Parafilm® until the isotopic analysis. The exhausted vials were successively recovered in 100 °C oven for 24 hours, while the capillary tubes were cleaned by acetone and then dried. All the plant samples were weighted before and after water extraction, and after the oven-drying at 100 °C for 24 hours to determine the extraction efficiency. We obtained an average extraction efficiency of 98.6 % (n = 65), whereas the median was 100 %.

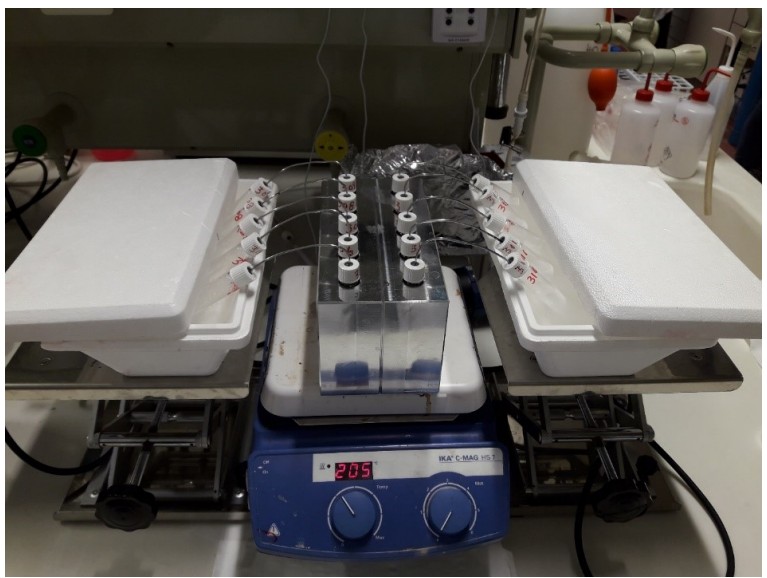

**Figure 3.** The cryogenic vacuum distillation (CVD) system at the Faculty of Science and Technology of the Free University of Bozen-Bolzano, based on the method developed by Koeniger et al. (2011).

### 3.3   Isotopic analysis

Isotopic analyses were performed by isotope-ratio mass spectrometry (IRMS) at the Faculty of Science and Technology of the Free University of Bozen-Bolzano. All water samples were analyzed using an IRMS (Delta V Advantage Conflo IV, Thermo Fisher Scientific Inc., Waltham, Massachusetts, USA), coupled with a Thermo Scientific Gas Bench II to determine $\delta^{18}O$.

For $\delta^{18}O$, water samples were placed in Exetainer® vials and the headspace flushed by a 0.3 % $CO^2$-He gas mixture of
known isotopic composition. After an equilibration phase of 24 hours, the headspace vapour phase was injected 8 times.

$\delta^2H$ was determined by direct injection of sample into the IRMS, through Thermo Scientific High Temperature Conversion
Elemental Analyzer (TC/EA, Thermo Fisher Scientific Inc., Waltham, Massachusetts, USA), equipped with an autosampler
(Thermo Scientific AI/AS 3000).

The samples were calibrated with standards relative to the Vienna Standard Mean Ocean Water. The standard deviation of
the isotopic measurements performed by the IRMS was 2.5 ‰ and 0.10 ‰ for $\delta^2H$ and $\delta^{18}O$, respectively.

## 3.4 Data analysis

The isotopic values of the samples were grouped based on the extraction method (i.e., SPC and CVD) and plant species (i.e.,
alder, apple, chestnut, and beech trees). In addition, samples extracted by CVD were grouped based on the collected plant tissue
(i.e., leaves (CVD-L), twig with bark (CVD-TwB), twig without bark (CVD-T), twig close to the trunk of the tree (CVD-TcT)
and wood core (CVD-WC)). In total, we considered 24 groups of samples for data analyses.

The samples were plotted in the dual-isotope space, together with the Local Meteoric Water Lines (LMWLs) of the three
study areas, obtained by the ordinary least squares regression (Marchina et al., 2020), to identify potential evaporated samples.
For each sample, we computed the line-conditioned excess (lc-excess*), which considers the deviation from the LMWL and
the uncertainty in the isotopic composition (Landwehr and Coplen, 2006), as follows:

$$lc - excess^* = \frac{\delta^2H - a \times \delta^{18}O - b}{S} \tag{1}$$

where *a* and *b* are the slope and the intercept of the LMWLs for each of the three study sites (equations reported in Fig. 4), and
*S* is the measurement uncertainty (Landwehr and Coplen, 2006). *S* was determined as follows:

$$S = \sqrt{SD_{\delta^2H}^2 + (a \times SD_{\delta^{18}O})^2} \tag{2}$$

where $SD_{\delta^2H}$ and $SD_{\delta^{18}O}$ are the typical standard deviations of the isotopic measurements (in our case, 2.5 ‰ and 0.10
‰ for $\delta^2H$ and $\delta^{18}O$, respectively). *S* resulted in 2.60, 2.61 and 2.63 in Ahr/Aurino, Laas/Lasa and Ressi, respectively. lc-
excess* values were used to investigate whether there was a marked offset of the samples from the LMWLs. Negative values
of lc-excess* mean that the samples experienced isotopic fractionation by evaporation or other fractionation processes (these
samples plot below the LMWL).

Scatter plots between SPC with CVD-T, CVD-TwB, CVD-TcT and CVD-WC samples were used to assess differences
(overestimation or underestimation) in the isotopic values. The Friedman repeated measures analysis of variance on ranks,
paired with a multiple comparison test based on the Tukey method, was used to identify significant differences (at the 0.05
level) in the isotopic composition, lc-excess* and volume of plant water extracted by the two methods and for the various
tissues, collected from alder and apple trees (these tests were not applied to chestnut and beech isotopic data because the paired
samples were < 4). The Welch two-sample t-test was used to assess whether the differences in the isotopic composition of SPC
and CVD-L samples from alder and apple trees were significant (at the 0.05 level).

To evaluate the differences in the isotopic composition between SPC and CVD extracted samples, while accounting for the uncertainty in the isotopic measurements, we computed the Z-scores for each paired sample and isotope (Wassenaar et al., 2012; Orlowski et al., 2016b), as follows:

$$Z - score = \frac{|CVD - SPC|}{SD} \tag{3}$$

where CVD is the $\delta^2$H or $\delta^{18}$O value of the cryogenic extracted samples, SPC is the $\delta^2$H or $\delta^{18}$O value of the SPC samples, and $SD$ is the typical standard deviation of the isotopic measurements. For the CVD samples, we distinguished the various plant tissues, i.e. CVD-T, CVD-TwB, CVD-TcT and CVD-WC. Similar to Orlowski et al. (2016b), for Z-score < 2, we considered the difference between the extraction methods acceptable (i.e., the observed difference can be considered equal or lower than the uncertainty in the isotopic measurements), for 2 < Z-score < 5, the difference was considered questionable, whereas for Z-score > 5 the difference was defined as unacceptable.

Scatter plots, the Friedman repeated measures analysis of variance on ranks (Scheff, 2016), and the Z-score analysis were applied only to those groups of samples that were not greatly affected by evaporation (i.e., CVD-L samples were not considered). We applied the Friedman repeated measures analysis of variance on ranks, instead of analysis of variance, because the repeated samples were few and non-normally distributed. The statistical analyses and the plots were prepared using SigmaPlot, Microsoft Excel and R.

## 4 Results

### 4.1 Isotopic variability across extraction methods and plant tissues

The volume of plant water extracted by the two methods and used for the isotopic analysis was significantly different (p < 0.001, Friedman repeated measures analysis of variance on ranks). Specifically, the volume of CVD-L samples was much larger compared to the volume of SPC samples (Table 1), and SPC samples had a significantly smaller volume compared to CVD-L, CVD-T and CVD-TwB samples (p < 0.05, pairs = 16, Tukey test run without accounting for species differentiation).

The isotopic composition of plant water varied considerably across the different plant tissues (Table 1 and Fig. 4). We found that CVD-L samples were more enriched in heavy isotopes than all the other plant tissues samples, and they plotted to the right side of the three LMWLs, highlighting a distinct evaporation signature (Fig. 4). Plant water extracted by SPC, CVD-T, CVD-TwB, CVD-WC and CVD-TcT generally plotted close to the LMWLs, except for three CVD-TwB samples from alder trees that were more depleted in heavy isotopes and plotted on the right side of the LMWL, and two samples from beech trees (one CVD-T and one CVD-TcT) that slightly plotted on the left side of the LMWL (Fig. 4). SPC samples were more enriched in heavy isotopes than CVD-T, CVD-TwB and CVD-WC samples collected in Ahr/Aurino and Laas/Lasa, whereas the differences between SPC and CVD samples (except for CVD-L) were less marked in Ressi, for both beech and chestnut trees (Fig. 4). The Welch two-sample t-test, applied only to alder and apple tree samples, showed that there was a significant difference in $\delta^2$H and $\delta^{18}$O of SPC and CVD-L samples (p < 0.001 for all four tests).

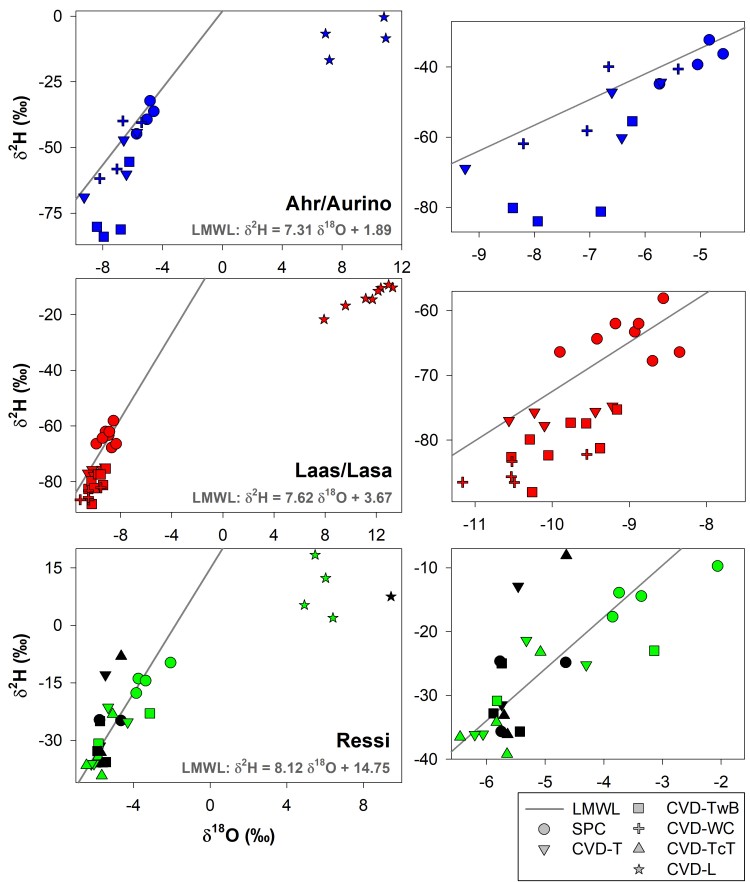

**Figure 4.** Dual-isotope plot for plant water samples extracted by Scholander-type pressure chamber (SPC), and cryogenic vacuum distillation (CVD) for different plant tissues (CVD-T, CVD-TwB, CVD-WC, CVD-TcT and CVD-L indicate samples extracted by CVD from twig without bark, twig with bark, wood core, twig close to the trunk of the tree and leaves, respectively) and species (alder, apple, chestnut and beech indicated in blue, red, green and black, respectively). Local Meteoric Water Lines (LMWLs) of the three study sites are plotted in gray: Ahr/Aurino: $\delta^2H = 7.31 \times \delta^{18}O + 1.89$, Laas/Lasa: $\delta^2H = 7.62 \times \delta^{18}O + 3.67$ (Penna et al., 2021), Ressi: $\delta^2H = 8.12 \times \delta^{18}O + 14.75$ (Marchina et al., 2020). The three plots on the right column represent a zoom in on SPC, CVD-T, CVD-TwB, CVD-WC and CVD-TcT samples.

**Table 1.** Sample size, median sample volume, median isotopic composition, and median lc-excess* of the samples extracted by Scholander-type pressure chamber (SPC) and cryogenic vacuum distillation (CVD) from different plant tissues (L: leaves; T: twig without bark; TwB: twig with bark; WC: wood core; TcT: twig close to the trunk), and species in the three study sites (Ahr/Aurino, Laas/Lasa and Ressi). Note that SPC samples consisted of lignified twigs with bark and leaves attached to the twig.

| Plant species | Sample type | Sample size | Median volume ($\mu$l) | Median $\delta^2$H (‰) | Median $\delta^{18}$O (‰) | Median lc-excess* (‰) |
|---|---|---|---|---|---|---|
| Alder (Ahr/Aurino) | SPC | 4 | 126 | -37.8 | -4.95 | -1.7 |
| | CVD-L | 4 | 950 | -7.6 | 8.99 | -29.2 |
| | CVD-T | 4 | 550 | -53.7 | -6.51 | -1.5 |
| | CVD-TwB | 4 | 550 | -80.7 | -7.37 | -9.3 |
| | CVD-WC | 4 | 100 | -49.4 | -6.86 | -1.3 |
| Apple (Laas/Lasa) | SPC | 8 | 108 | -63.8 | -8.91 | 1.0 |
| | CVD-L | 8 | 1910 | -12.9 | 11.92 | -41.1 |
| | CVD-T | 5 | 700 | -75.7 | -10.10 | -1.7 |
| | CVD-TwB | 8 | 710 | -80.6 | -9.91 | -3.3 |
| | CVD-WC | 5 | 280 | -85.7 | -10.52 | -3.5 |
| Chestnut (Ressi) | SPC | 4 | 196 | -14.2 | -3.55 | -0.6 |
| | CVD-L | 4 | 1550 | 8.8 | 5.76 | -19.2 |
| | CVD-T | 4 | 950 | -30.6 | -5.69 | -0.4 |
| | CVD-TwB | 2 | 1050 | -26.9 | -4.48 | -2.0 |
| | CVD-TcT | 4 | 1300 | -35.4 | -5.74 | -0.1 |
| Beech (Ressi) | SPC | 3 | 198 | -24.8 | -5.75 | -0.7 |
| | CVD-L | 3 (1 for $\delta^2 H$) | 800 | 7.5 | 4.87 | -31.9 |
| | CVD-T | 3 (2 for $\delta^2 H$) | 800 | -22.2 | -5.65 | 3.2 |
| | CVD-TwB | 3 | 700 | -32.8 | -5.74 | 0.1 |
| | CVD-TcT | 3 | 1300 | -33.1 | -5.64 | -0.6 |

The relation between $\delta^2$H and $\delta^{18}$O of plant water extracted by SPC and CVD showed differences among plant tissues and the four species (Fig. 5 and 6). Indeed, we observed that most of the samples did not plot on the 1:1 line, and there were very large differences between $\delta^2$H of SPC and CVD-TwB, particularly for alder tree samples (the absolute differences varied between 16.1 and 48.9 ‰) and apple tree samples (the absolute differences varied between 12.0 and 21.7 ‰) (Fig. 5b). For alder, apple and chestnut tree samples, we found that $\delta^2$H of SPC was always more positive than $\delta^2$H of CVD samples, except for CVD-L (Fig. 4). The $\delta^2$H of plant water collected from beech trees by CVD-T, CVD-TwB and CVD-TcT was not systematically more enriched or depleted than $\delta^2$H of SPC samples.

Likewise, we found differences in $\delta^{18}$O values between SPC and CVD samples (Fig. 6). However, compared to $\delta^2$H, more samples plotted closer to the 1:1 line. The differences between SPC with CVD-T and CVD-TwB of beech samples were small

(the median of the absolute differences was 0.22 ‰, n = 6), and the samples plotted very close to the 1:1 line (Fig. 6a,b). SPC samples from alder, apple and chestnut trees were less negative in $\delta^{18}O$ than CVD samples, but for apple tree samples the differences between SPC and CVD-TwB were relatively small (the median of the absolute differences was 0.57 ‰, n = 8).

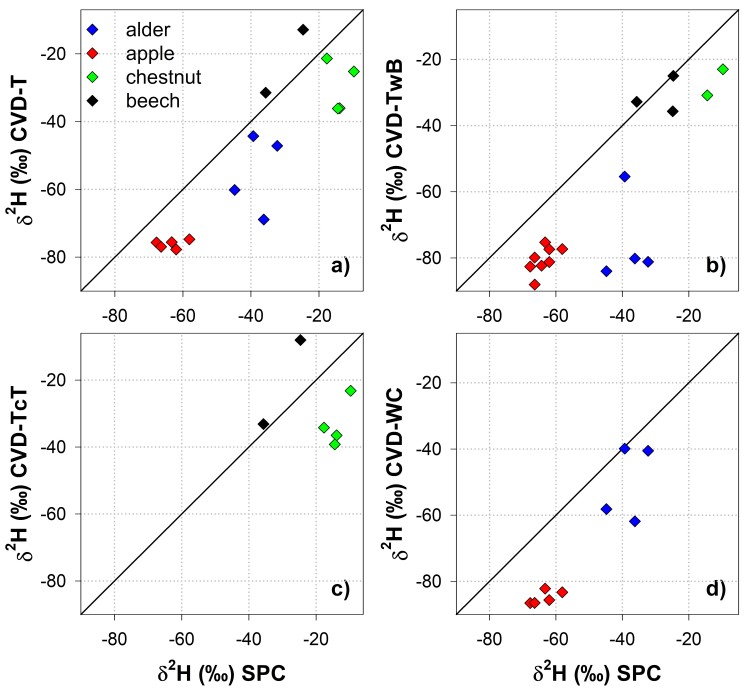

**Figure 5.** Relation between $\delta^2H$ values in plant water extracted by SPC (i.e., Scholander-type pressure chamber) and CVD (i.e., cryogenic vacuum distillation), grouped by plant tissue and species. The solid black lines represent y = x.

## 4.2 Effect of the extraction method on plant water isotopic composition

The Friedman repeated measures analysis of variance on ranks (applied only on alder and apple isotopic data) showed that there was a significant effect (with $\alpha = 0.05$) of the extraction method and plant tissue on $\delta^2H$ and $\delta^{18}O$ of plant water (Fig. 7). For alder trees, we found that SPC samples were significantly different in $\delta^2H$ and $\delta^{18}O$ from CVD-TwB samples (p < 0.05, pairs = 4, Tukey test). For apple trees, SPC samples differed in $\delta^2H$ and $\delta^{18}O$ from CVD-WC samples (p < 0.05, pairs = 5, Tukey test).

We observed that there was not a significant effect of the extraction method on lc-excess* (p > 0.05, Friedman repeated measures analysis of variance on ranks), except for CVD-L samples (median lc-excess* was always very negative; Table 1). Besides CVD-L samples, a marked negative lc-excess* (larger than the uncertainty in the isotopic measurements) was found only for CVD-TwB samples collected from alder trees. Median lc-excess* was close to zero for chestnut and beech tree samples (CVD-T of beech samples even had positive values), as well as for some alder (SPC, CVD-T and CVD-WC) and apple (SPC

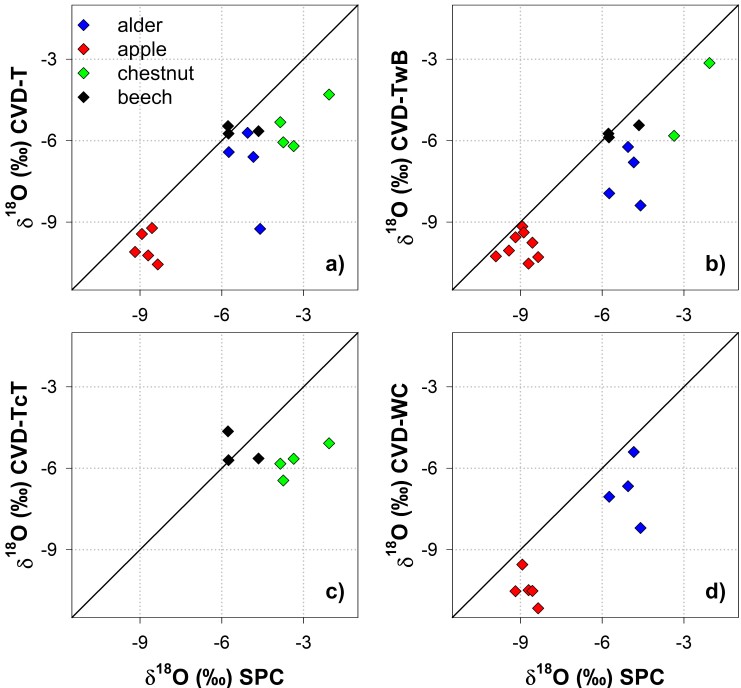

**Figure 6.** Relation between $\delta^{18}O$ in plant water extracted by SPC (i.e., Scholander-type pressure chamber) and CVD (i.e., cryogenic vacuum distillation), grouped by plant tissue and species. The solid black lines represent y = x.

and CVD-T) tree samples (Fig. 7 and Table 1). Interestingly, SPC samples from apple trees had a positive median lc-excess* compared to the negative lc-excess*, and larger offset from the LMWL of CVD-TwB and CVD-WC samples collected from the same plants (Fig. 7 and Table 1).

### 4.3 Are the differences between SPC and CVD larger than the uncertainty in the isotopic measurements?

The Z-score analysis showed that generally the differences between $\delta^2H$ and $\delta^{18}O$ of SPC and CVD samples were larger than the uncertainty in the isotopic measurements (Fig. 8). Due to the larger uncertainty in $\delta^2H$ measurements compared to $\delta^{18}O$ (based on the IRMS used in this study), we observed that the computed Z-scores were smaller for $\delta^2H$ than for $\delta^{18}O$.

For $\delta^2H$, Z-scores varied between 0.1 (computed between SPC and CVD-TwB for samples collected from a beech tree in Ressi) and 19.6 (computed between SPC and CVD-TwB for samples collected from an alder tree in Ahr/Aurino). The median Z-scores for $\delta^2H$ were 6.0, 6.5, 6.6 and 7.6 computed between SPC with CVD-T, CVD-TwB, CVD-TcT and CVD-WC, respectively; these median values indicate that more than 50 % of the Z-scores were above the limit for questionable differences (Z-score = 5) between the extraction methods (Fig. 8a). For $\delta^2H$, only few Z-scores (about 10 %) were lower than the upper limit for acceptable differences (Z-score = 2) between the methods. Overall, the smallest differences in $\delta^2H$ (and Z-scores) were found between SPC and CVD-T, followed by SPC and CVD-TwB (Fig. 8a).

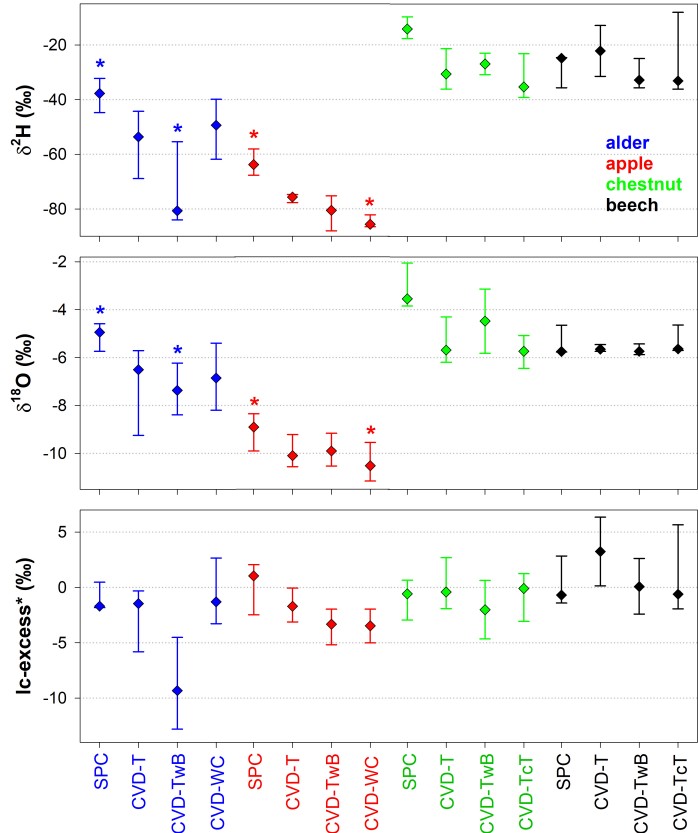

**Figure 7.** Median isotopic composition and lc-excess* of plant water extracted by SPC (i.e., Scholander-type pressure chamber) and CVD (i.e., cryogenic vacuum distillation), grouped by plant tissue and species. Error bars represent the minimum and the maximum values. Asterisks above the dots indicate significantly different groups ($p < 0.05$, multiple comparison test based on Tukey method, run after the Friedman repeated measures analysis of variance on ranks).

For $\delta^{18}O$, Z-scores varied between 0.1 (computed between SPC and CVD-T for samples collected from a beech tree in
Ressi) and 46.6 (computed between SPC and CVD-T for samples collected from an alder tree in Ahr/Aurino). The median Z-scores for $\delta^{18}O$ were 12.4, 10.8, 19.8 and 16.1 computed between SPC with CVD-T, CVD-TwB, CVD-TcT and CVD-WC, respectively; these results indicate that about 75 % of the Z-scores were above the limit for questionable differences between the extraction methods (Fig. 8b). For $\delta^{18}O$, only few Z-scores (less than 10 %) were lower than the upper limit for acceptable differences between the methods. The smallest differences in $\delta^{18}O$ (and Z-scores) were observed between SPC and CVD-TwB,
followed by SPC and CVD-T (Fig. 8b).

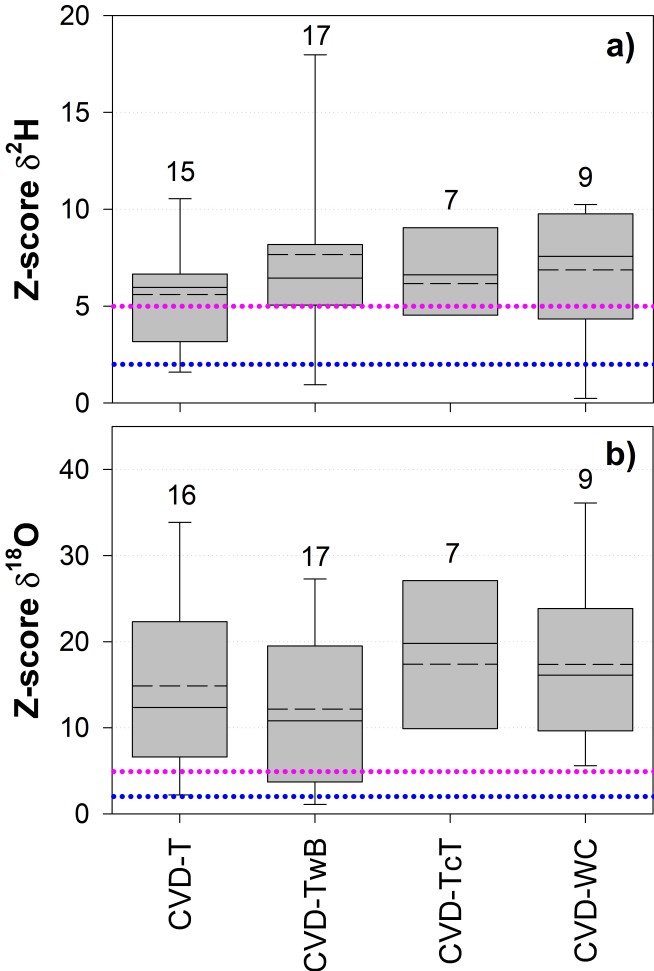

**Figure 8.** Dimensionless Z-score values for $\delta^2$H (a) and $\delta^{18}$O (b) grouped by sample types (CVD: cryogenic vacuum distillation; T: twig without bark; TwB: twig with bark; TcT: twig close to the trunk; WC: wood core). Samples from the four species were grouped together, and numbers above the boxes represent the sample size. The boxes indicate the $25^{th}$ and $75^{th}$ percentile, the whiskers indicate the $10^{th}$ and $90^{th}$ percentile, whereas the horizontal solid and dashed lines within the box mark the median and the mean, respectively. The dotted blue and pink lines represent the upper limits for acceptable (Z-score = 2) and questionable (Z-score = 5) differences, respectively, between the SPC (i.e., Scholander-type pressure chamber) and the CVD extracted samples.

## 5    Discussion

### 5.1    Advantages and limitations of water extraction by SPC

The SPC has the advantage of extracting plant water that is likely used for transpiration and it is not tightly stored in the plant tissues (Meiri et al., 1975; Grossiord et al., 2017). The water extraction by SPC can be applied directly in the field or in a laboratory after a proper handling and transport of the vegetation material in sealed bags. The procedure for the extraction of plant water is also simple because it does not require specific laboratory work (such as, handling liquid nitrogen and transferring samples to different vials), as for the CVD system. In addition, water extraction by SPC generally lasts few minutes depending on the plant water potential, whereas the extraction by CVD could last from few minutes (15 min in this study) up to hours (Millar et al., 2018). The easy and fast application (without extensive laboratory work) of the SPC for plant water extraction can be considered comparable to the simple and low-cost methods developed by Fischer et al. (2019).

Despite the advantages listed above, our sampling approach showed that water extraction by SPC is not always satisfactory in terms of sampling volume and extraction times. For instance, for some twig samples collected in Ressi during a dry period in July 2017, we had to apply a 3.0-MPa pressure for the extraction of at least 60 $\mu$l for the isotopic analysis by IRMS, and the whole sample extraction lasted about 10 min. The sampling procedure was also complicated by the extraction of few small water droplets and air bubbles that were difficult to trap into the vials. Conversely, the plant water extraction by CVD was performed for all the samples, and generally obtaining sampling volumes much larger than 100 $\mu$l. Furthermore, plant water extracts obtained by SPC usually were darker (yellowish or even brownish) compared to water extracts by CVD. The dark colour of the SPC plant water extracts suggests a possibly large concentration in organic compounds (Millar et al., 2018), likely due to a partial destruction of plant cell walls, and/or a potential contamination by the phloem (we did not peel this tissue from the SPC samples). In our case, the sampling volume was not enough to quantify the concentration of organic compounds. Compared to this study, Gei$\beta$ler et al. (2019) performed a post-processing analysis (by a spectral contamination identifier software) to quantify the spectral contamination of organic compounds, and found that for six stem water samples from *Acacia mellifera* shrubs the relative degree of interference from contaminants in the extracted water was clearly higher for CVD than SPC (Fig. S1 in Gei$\beta$ler et al. (2019)).

### 5.2    Difference between the two techniques and implications for ecohydrological studies

Our results showed that twig samples (with leaves) obtained by SPC, differently from the CVD-L samples, did not show any offset compared to the LMWL of each site (Fig. 4). This might suggest that we did not extract significant volumes of leaf water, which is typically subject to water-vapour exchanges with the low atmosphere (e.g., Cernusak et al., 2016; Benettin et al., 2021). The lc-excess* values of plant water extracted by SPC and CVD were generally close to zero (except for CVD-L, and CVD-TwB samples collected from alder trees, Fig. 7), indicating a limited deviation from LMWLs (generally smaller than the uncertainty in the isotopic measurements; Table 1 and Fig. 7), with even positive values for plant water samples extracted from some beech trees in Ressi. This observation indicates that either the applied methods did not alter the isotopic composition of plant water during the extraction and did not determine a marked deviation from the LMWLs, if any isotopic fractionation

occurred during the water transport from roots to leaves, or that both methods did not access plant water with a significant offset from the LMWLs. Despite the similarity in lc-excess* between samples extracted with the two methods (no significant differences were found by the p > 0.05, Fig. 7), we observed that plant water collected by CVD method from alder, apple, and chestnut trees was always more depleted in heavy isotopes (both $\delta^2$H and $\delta^{18}$O), and in some cases significantly different from plant water samples extracted by SPC (Figs. 5, 6 and 7). Our results are partly in contrast with the findings of Barbeta et al. (2022), who found a marked depletion in $\delta^2$H of plant water extracted by CVD compared to xylem water obtained by the cavitron technique, whereas we observed an offset for both isotopes. As expected, given the different plant water domain accessed by the two methods, the water extracted by SPC and CVD showed differences in the isotopic composition among plant tissues, larger than the uncertainty in the isotopic measurements, and such differences were considered unacceptable in terms of Z-scores (Fig. 8). As expected, these results are in contrast with those found by Geiβler et al. (2019), who reported a large variability in $\delta^{18}$O, but no statistical differences for six stem water samples of *Acacia mellifera* extracted by CVD and SPC. However, we must consider that Geiβler et al. (2019) compared a limited number of samples and a different species, and used samples deprived of the phloem tissue.

In our study, we relate the observed isotopic differences between SPC and CVD samples to various factors, such as the possible effect of organic compounds on the isotopic composition of plant water (although we did not check this effect) and/or the plant water domain accessed by each method (e.g., more mobile plant water extracted by SPC vs. all plant water extracted by CVD). Millar et al. (2018) reported that different methods can extract different water domains within the plants, and CVD extracts up to 99 % of the water in a sample, i.e., CVD accesses total plant water. Conversely, SPC mainly extracts water present in the xylem conduits, and given the much smaller sample volumes we collected by SPC than by CVD (on average $\approx$ 135 $\mu$l for SPC vs. $\approx$ 955 $\mu$l for CVD, Table 1), we likely accessed different plant water domains when using the two methods, with SPC pulling out more easily mobile water (i.e., xylem and inter-cellular water) than water stored in living cells. Even for CVD-TcT samples, which were supposed to have older tissues and more dead cells than CVD-T and CVD-TwB, we need to consider that CVD could still extract significant water amounts stored in living xylem parenchyma cells, and the total ray and axial parenchyma tissue fractions can be 21.1 $\pm$ 7.9 % (average $\pm$ standard deviation) in temperate angiosperm trees (Morris et al., 2016). Water taken up by roots and transported in the xylem conduits can reach the leaves very rapidly and be available for transpiration, whereas the living cells (which are abundant in the leaves and other non-lignified tissues) may store water taken up several days or even weeks before the sampling date (Sprenger et al., 2019). Therefore, water taken up by roots in different periods and stored in different tissues might have very different isotopic compositions, something that both methods cannot clearly distinguish.

The ability of SPC and CVD to extract different plant water domains has implications for studies investigating the water sources exploited by plants for transpiration (e.g., Zhao et al., 2016; Barbeta et al., 2019; Allen and Kirchner, 2022; Barbeta et al., 2022). Indeed, significantly different isotopic compositions in the extracted plant water, which can also be related to the $\delta^2$H bias observed in the CVD extraction in recent studies by Chen et al. (2020) and Barbeta et al. (2022), can complicate the identification of the water sources contributing to transpiration, and can result even in substantially different estimations of the contributing water sources (Barbeta et al., 2019; Allen and Kirchner, 2022). In this view, such ecohydrological studies

should rely more on methods extracting apoplastic water representative of transpiration (like SPC) than on methods extracting, in addition to xylem water, also other plant water fractions (stored in living cells) that are likely much older (even decades, as reported in Zhang et al. (2017)) than the actual xylem water. In addition, possible exchanges during transportation among different plant water domains would result in mixing of water having different ages, and likely different isotopic composition (Ellsworth and Williams, 2007; Zhao et al., 2016).

Furthermore, our results show that the differences in the isotopic composition between SPC and CVD vary not only based on the plant tissue used for CVD, but also based on the plant species (Figs. 5, 6 and 7). Given the isotopic differences among various species and the results obtained by Gei$\beta$ler et al. (2019), more research is needed to compare multiple extraction methods (SPC should not only be compared to CVD, but as well as to direct vapour equilibration, microwave extraction, centrifugation, cavitron flow-rotor etc., that might access different plant water domains). Future inter-method comparisons should be carried out across various environments and plant species to investigate the technical and physical factors possibly altering the isotopic composition of plant water during the extraction and/or potentially different plant water domains accessed by each method, in order to elaborate standard protocols for ecohydrological research relying on the isotopic signature of plant water.

## 5.3 Limitations of this study

Our results contribute to the pressing need of comparing different plant water extraction techniques to understand which plant water domains are accessed by different methods (Penna et al., 2018). Despite the importance of our findings for the isotope ecohydrological community, we acknowledge some limitations in the experimental setup, which may impact the interpretation of the results.

Firstly, our experiment was not designed to test whether plant water extracted by SPC from twigs with or without bark had a significantly different isotopic composition. Contrary to Gei$\beta$ler et al. (2019), who performed their experiment after we performed ours, we did not remove bark and the leaves attached to the twig (similarly to Penna et al. (2013, 2021)). However, we found no direct influence of leaf water isotopic composition on our SPC samples (no deviation from LMWLs, see Fig. 4), and therefore, we could compare plant water extracted by SPC to plant water obtained by CVD-TwB. Nonetheless, our results are not directly comparable to the findings by Gei$\beta$ler et al. (2019), and future research should aim to test whether SPC is able to extract phloem from twigs with bark and green tissues, and whether there is a significant isotopic difference with SPC samples obtained from twigs without bark.

Secondly, our experimental scheme did not include the quantification of organic compounds, and the water volume obtained for SPC samples was not enough to carry out such analyses. The quantification of organic compounds would have been important to determine whether the destruction of plant cell walls in SPC samples could increase the number of organic contaminants in the extracted water, as well as alter the isotopic composition of the samples, and to quantify and compare the type and amount of organic compounds in SPC and CVD samples. Given that we were not able to quantify the concentration of organic compounds in our samples, and Gei$\beta$ler et al. (2019) carried out a quantification of the spectral contamination

for just six samples and only one plant species, we recommend that future inter-method studies should compare the isotopic composition of plant water extracted by SPC and CVD, as well as the measured concentration of organic compounds.

Another limitation of our experimental setup is represented by the sampling time which differed, due to logistic constraints,
at the three study sites. Indeed, we cannot exclude a variability in the daily dehydration-rehydration cycles inside the stems related to the phloem-xylem water transfer (Pfautsch et al. , 2015) during the two sampling times considered in this study (i.e., after the sunset and at the sunrise). A different phase in the phloem-xylem water transfer might have determined a different contamination of phloem during the SPC extraction at the three sites. Therefore, future inter-method studies should try to minimize the difference in sampling times for the collection of the various samples, or design experiments which consider
multiple sampling campaigns during the day, and the daily variation in the phloem-xylem water transfer as a factor potentially affecting the isotopic composition of the plant water.

Finally, based on our experimental setup, we were not able to determine exactly which plant water domain was accessed by SPC. In terms of sample type, our SPC samples were more comparable to CVD-TwB samples because we did not remove bark, and our results confirmed that there was no influence of evaporated leaf water on the SPC samples. Therefore, similarly
to CVD-TwB, we cannot exclude that our SPC samples were contaminated by the extraction of water from the phloem tissue. However, the smaller sample volumes we collected by SPC compared to CVD (Table 1), and the different isotopic composition, particularly shown by the Z-score analysis (Fig. 8), indicate that SPC tended to access more easily the mobile xylem water and inter-cellular water than the less mobile intra-cellular, cell wall, and organelle constrained water (Millar et al., 2018). Despite this, our study did not resolve whether SPC was able to extract all the less mobile plant water (besides likely cell walls), and
whether the results were affected by other factors, such as the presence of organic compounds or the sampling time. Given that we were not able to determine exactly the plant water domains accessed by SPC, future comparison studies between the SPC and CVD techniques should carefully consider the sample types (both with bark and without bark to assess whether SPC extracts phloem), the quantification of organic compounds, and the extraction of plant water using different external pressures. By applying different external pressures to plants not suffering from high water deficit, and under the assumption that more
and less mobile plant water have different isotopic compositions, it may be possible to test whether SPC can extract mobile plant water when a low external pressure is applied, and a mixture of more and less mobile plant water during the application of higher pressures.

## 6   Concluding remarks

In this work, we analyzed the variability in the isotopic composition of plant water extracted by cryogenic vacuum distillation
(CVD) and Scholander-type pressure chamber (SPC), also considering the potential variability in the isotopic signature of the plant water extracted from various tissues by CVD (i.e., leaves, twig without bark, twig with bark, twig close to the trunk of the tree, and wood core), and from different plant species (i.e., alder, apple, chestnut and beech). The procedure for the extraction of plant water by SPC is simple, can be carried out in the field, and it does not require specific laboratory work as in case of CVD. However, the main limitation of SPC is the very small water volume that can be extracted from the lignified twig

during water deficit conditions, compared to CVD. Moreover, our results indicated that the isotopic composition of plant water extracted by SPC and CVD was significantly different. While SPC and most of the CVD samples (except for CVD applied to leaves) did not exhibit an evaporative signature, there was a large isotopic variability among the samples. We found that, for beech tree samples, the difference in both $\delta^2$H and $\delta^{18}$O obtained by the two extraction methods was smaller compared to the difference observed for alder, apple and chestnut tree samples. Specifically, the isotopic composition of alder, apple and

chestnut plant water extracted by SPC was more enriched in heavy isotopes compared to samples obtained by CVD applied to twigs or wood cores. Based on these results, we conclude that SPC accesses only the more mobile part of the plant water fraction that CVD does, and therefore, SPC is not an alternative to CVD in terms of plant water accessed. Therefore, studies aiming to quantify the relative contribution of the water sources to transpiration should rely more on the isotopic composition of xylem water transpiring at the moment of the sampling or during the sampling day (which is theoretically sampled by SPC),

than the isotopic composition of total plant water (sampled by CVD), which also contains a fraction of water that could be stored in plant tissues for a longer time. Based on our findings, we call for future research investigating the same methods across more plant species, and quantifying the organic compounds in both SPC and CVD samples to determine the effect on the isotopic composition of plant water.

*Data availability.* Data are available from the corresponding author upon reasonable request.

*Author contributions.* DP conceptualized the methodological comparison between the two methods. JF, GZ and ME designed the research. JF and ME collected field data in the Ahr/Aurino and Laas/Lasa sites, whereas CM, GZ and AA carried out the field campaigns in the Ressi catchment. JF and AA performed the plant water extraction by the cryogenic vacuum distillation. FS and DZ provided support for the sampling in Laas/Lasa and the laboratory activities. VC and TA provided technical support and comments for the plant water extraction by the Scholander-type pressure chamber. GZ analyzed the data set, and prepared the first draft of the manuscript with contributions from AA,

JF, ME and CM. All the authors contributed to the editing of the manuscript. FC, MB and MT funded the research.

*Competing interests.* The authors declare that they have no conflict of interest.

*Acknowledgements.* GZ and CM acknowledge the financial support provided by Fondazione Cassa di Risparmio di Padova e Rovigo (Italy) (research project "Ecohydrological Dynamics and Water Pathways in Forested Catchments", Bando Starting Grants 2015). Research in the Ressi catchment was also supported by the Italian MIUR Project (PRIN 2017) "WATer mixing in the critical ZONe: observations

and predictions under environmental changes-WATZON" (national coordinator: Marco Borga). Research in the Ahr/Aurino study area was supported by the RIVERMOOD project funded by the Free University of Bozen-Bolzano (Italy). Research in the Laas/Lasa study site was supported by the project "Parco Tecnologico-Tecnologie Ambientali" of the Autonomous Province of Bozen-Bolzano (Italy).

The authors would like to thank Christian Ceccon for the isotopic analyses and the support during the laboratory activities at the Free University of Bozen-Bolzano (Italy).

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
