# Peer review of "A comparative study of plant water extraction methods for isotopic analyses: Scholander-type pressure chamber vs. cryogenic vacuum distillation"

_Hydrology and Earth System Sciences, 2020_

## Referee Comment (RC1) · Anonymous Referee #1 · 18 Nov 2020

This paper provides further examination of a promising plant water extraction method for isotopic analyses, Scholander-type pressure chamber water extraction (SPC), and compares results of that method to the conventional plant water extraction method of cryogenic vacuum distillation (CVD). This comparison is necessary to determine possible advantages, disadvantages, and implicit assumptions related to method choice for many ecohydrologic studies. However, I have what I think are valid concerns about the current version.

General feedback:

[Figure]

As mentioned in the introduction, there is an overarching issue within the ecohydrological community where differences in the isotopic composition of plant water have been related to extraction methods. These differences have been attributed to either one or a combination of the following: 1) Inter-lab differences: lab specific protocols, setups, accuracies, efficiencies 2) Inter-method differences: various alterations and various accessibility 2a) alterations of plant material associated with specific methods: e.g., fractionation effects associated with incomplete extractions or specific extraction methods, co-extraction of organic substances 2b) variability in the proportion of plant water domains accessed via different methods: lower residence time domain from xylem tracheary elements (dead cells) participating in transportation of water from soil to leaves versus higher residence time domain of living cells not participating in transportation of water. (I use domains rather than pools since that is the language recently being used for residence time of soil water in ecohydrology. Pools and domains are likely interchangeable and I am not saying one should be used over another, just that the use should be consistent throughout the paper. I will refer to them as domains for the remainder of this review to be consistent.)

For this paper the authors focus on (2a) with d-excess analyses, but there are issues (highlighted in specific comments) with whether d-excess can exclude considerations of (2b). Additionally, (2b) is poorly discussed for samples and analyses throughout paper. There are also some study design limitations for (2b) which would be hard to address by including more samples since sampling conditions will not be consistent, but the limitations should be mentioned for any future studies to address.

The authors argue that the SPC method needs more investigation into its merits for ecohydrologic studies and that the SPC results need to be compared to conventional method of CVD. Hence, the authors two objectives (L79). However, there are issues with how they can address the first objective because their study design does not have a direct comparison between methods since the sampling materials are not the same. Granted a shoot and a twig are similar, if not the same in some cases, but the use

of these different terms is confusing from the onset. Furthermore, SPC samples are from one year old shoots with leaves and bark intact whereas CVD samples are from multiple tissue types that make up elements in a one year old shoot, but these elements are never all together (there are no CVD samples of one year old shoots with leaves and bark included).

These differences in tissue type for each method seem to be part of the reason for the second objective, but there is inadequate considerations of the (2b): variable access issues. With leaves still intact for SPC extraction, the possibility of extracting leaf water from predominantly xylem conducting cells in leaves seems hard to discount, especially when authors state that water extraction of shoots via SPC ended when they had collected all the water flowing out of the shoots (L139). In addition, with bark still intact at collection site via SPC it is possible that phloem substrates are co-extracted, which is not mentioned when addressing how SPC samples were colorful in discussion(L285). In fact, I am perplexed when they cite Geißler et al. (2019) thoroughly in the introduction yet do not follow the same precautions, removing leaves and bark near collection site, to eliminate the contribution of water from leaves and phloem from live tissues still intact near collection site(L131). In methods section 3.2, authors state that "Samples with and without bark were used to test whether the plant water extracted by the SPC method had an isotopic composition more similar to CVD-extracted bulk plant water (i.e., CVD-TwB) or to plant water deprived of phloem tissues (i.e., CVD-T)". This somewhat states the assumptions by the authors that a twigs with bark have more living cells, which is reasonable. Furthermore, twigs without bark are predominantly dead cells and isotopic compositions of CVD-T samples should be more similar to isotopic composition of SPC samples (under the logic stated in introduction that SPC is collecting water primarily from dead cells). Although this logic holds up, this instance and other areas that discuss lignified tissues are missing the necessary consideration that total extractions of a twigs without bark via CVD could still be extracting water from living xylem parenchyma cells in the complex woody tissue. The proportion of xylem parenchyma varies by species and a recent global synthesis showed that the combined

ray and axial parenchyma content for angiosperm trees and shrubs averaged 26.3 ± 12.4% (Morris et al., 2016). A possible contribution of a quarter of the water extracted seems like an important consideration to me, especially as they argue the possibility of the living cells having very different isotopic compositions than water being conducted in xylem conduits (L332). Granted some details of variation in dead cells are included in description of CVD-TcT (L153), but the relative amount of living cells needs to be highlighted more.

Overall, the authors need to discuss limitations in their design and provide arguments of why they can directly compare samples for first objective (e.g., SPC vs CVD-T). In addition, they need to have a more thorough discussion about the variability of accessing plant water domains with each method, tissue type, and species. In regard to methods, this includes many instances of when they compare to other studies investigating differences between methods and provide inadequate context to what plant water domains are accessed for the comparison. Finally, there needs to be greater clarity in the paper with the use of the word "effects" on isotopic composition of plant water. The paper lacks necessary discussion distinguishing the possible effects that could alter plant material versus effects of accessing different plant material when collecting a water sample, particularly when assessing results and conveying the future directions.

Specific comments:

L5: could benefit from more explicit mention of what plant water domain SPC is accessing similar to that done by CVD. In other words, why does SPC need more attention other than it being rarely applied?

L9: kind of misleading that SPC was done on multiple separate plant tissues when it was done from a one year old shoot comprised of multiple plant tissues. Certainly does not have to be here, but in the discussion it would be helpful to recognize past work on variability of isotopic composition of multiple plant materials done by Zhao et al. (2016)

"Significant Difference in Hydrogen Isotope Composition Between Xylem and Tissue Water in Populus Euphratica"

L18: granted you preface with likely, but how do you know that pressure applied via SPC is not affecting any living cells and only dead cells? Pressure is applied through leaves, so aren't some living cells impacted, albeit minorly (especially considering your differences in CVD-L and SPC)? in your results you suggest cell walls were broken leading to co-extraction of organic substances by SPC.

L35: mentioning these ecohydrologic studies is necessary, but your introduction currently lacks the broader context of issues within the community. CVD has been the conventional method for so long with assumptions that root water uptake is non-fractionating process for the most part and that total plant water via CVD has been considered representative of transpiration due to water in plant tissue assumed to be in equilibrium or well-mixed. The well-mixed assumption has been recently questioned by ecohydrologic separation studies (many of which you cite here) and discussion around plant water domains similar to soil water domains. For instance, you mention Barbeta et al. (2019) in the end of the discussion, but I feel like it would also help to include some of the ideas/issues from that paper here to help preface why SPC should be investigated more.

L38: in addition to not altering plant material, these studies also want isotopic composition of plant water that is representative of transpiration which requires techniques that don't alter plant material and involves criticizing what plant water is accessed with each method (water from live versus dead cells).

L58: Millar et al. (2018) do categorize methods by what plant water domains(or pools in their case) are accessed and being more explicit about that here would help provide context to readers of why SPC is important to further investigate.

L61: what plant water domains were accessed with Fischer et al. (2019) various methods?

L71: what about the work by Ellsworth and Williams, 2007–"Hydrogen isotope fraction-ation during water uptake by woody xerophytes" which used modified SPC chamber to understand possible fractionation during root water uptake

L75: This mentioned comparison by Geißler et al. (2019) does have important nuances of how they performed SPC method and the sample material preparation.

L93: what is the rationale in after sunset? Why not when water deficit is lower, i.e., pre-dawn to mid-morning, especially if there are issues in acquiring enough volume via SPC? "at the downstream in the Ahr/Aurino study area" is unclear – photo of site has two sites with one more downstream than the other which is also much closer to the stream.

Table 1: unclear what plant tissue/material was used for SPC, adding this detail to the table description would be helpful

L129: what position in canopy were one year old shoots taken from? Range in size of diameter extracted like that mentioned in 3.2 for CVD-T and CVD-TwB (L146)? Was canopy position and aspect similar for all sample types being more closely compared (SPC, CVD-T, CVD-TwB)?

L131: what is the rationale in leaving the leaves on and bark intact near collection site? This does not follow rationale and guidance of Geißler et al. (2019) use of the SPC method to collect xylem water. Also, figure 2 has several leaves on the shoot in the chamber making the statement of "one or more leaves sealed inside chamber" somewhat unclear and added clarity would be beneficial.

L136: directly is a little vague, how does that differ from pipettes? Was the pressure chamber put on its side so that water could fall into vials directly via the help of gravity?

L139: rough timeframe? Longer or varied exposure to air/evaporation could impact comparison of results. All water implies that some water from leaves is incorporated in sample alongside shoot water which impacts how direct the comparisons can be

between SPC and CVD-L/CVD-T/CVD-TwB

L148: what about living ray and axial parenchyma cells in xylem? similar or related reasoning for leaves is lacking in L145.

L150: was phloem removed for wood core samples? How much of active xylem/sapwood was used in wood core and did wood core include heartwood?

L186: there are two additional LMWL sources missing here that are mentioned in figure 4 caption

L192: if you are trying to address whether method is fractionating water (or if you are accessing fractionated water) why not also do site specific lc-excess to more directly account for the local inputs? Especially with Ressi having very different LMWL this seems hard to gain much from the comparison with d-excess.

L195: what do you mean by "effects" ? Are the effects of the method altering isotopic composition? Or are they accessing different plant water domains? Or possibly both?

L231: Not true for CVD-L samples. In general, CVD-L samples seem to be poorly integrated in discussion and results. I believe that their differences could help highlight that SPC with leaves attached are accessing different plant water domain in leaf than domains accessed by CVD for CVD-L.

Figure 4: I think it would be helpful to include the GMWL on these plots to aid in visualizing how d-excess values are generated for each site.

L277: does this include comparable in plant water domains extracted?

L278: no specific mention of sampling times for each site/species sampled before this statement. Recommend putting SPC extraction times in methods

L286: what about phloem contributions via SPC extraction in addition to destruction of plant cell walls? According to methods the bark near the cut surface of one-year old shoot was not removed to limit contribution of phloem to extracted "water". What about

contributions from water transporting and non-water transporting leaf cells? So, does your use of SPC method contradict opening statement of the discussion by breaking cell walls? Or does the contribution of phloem need greater consideration with your use of SPC method?

L298: based off introduction of CVD and SPC accessing different amount of plant water, is it really a "drawback" that they aren't comparable? I guess it is unclear if you are trying to state the SPC is an alternative for simplicity or if you are also considering SPC as an alternative for accessing water representative of transpiration vs total water via CVD?

L301: consider revising this sentence to be more clear, particularly the section starting in "indicating" is confusing as is. At a broader scope, being that leaves were still attached, then does this support that SPC is not accessing the evaporatively enriched pool of water in leaves that is found in the living leaf cells?

L305: are you stating that the methods didn't fractionate plant water or that you didn't access fractionated water via both methods? Or both? I think this needs to be more clear for readers.

L306: I think it would be beneficial to point out that these values for Ressi are high with respect to other sites because the LMWL of Ressi plots above the GMWL in dual isotope space.

L313: what plant water is sampled (predominantly transporting cells or non-transporting cells?) via direct vapor equilibration and microwave extraction and how does that help contextualize the comparisons of SPC results to CVD results? Although Zhao et al. (2018) attributed results to only fractionation effects rather than additionally considering access of different plant water domains, they also reported similar more negative d2H values for CVD-stem, CVD-core, CVD-root compared to d2H values of xylem sap sampled via needles.

L321: "sampling material" alone lacks distinction of different plant water domains and how the proportion of those domains possibly vary for each sampling material used

L332: Mask? Isn't the possibility of accessing different plant water domains a reason for comparison between methods and your objectives? It seems to me that the study design is what precludes you from a more direct comparison (sampling material not the exact same for both methods).

L342: any future directions on why there may be differences between species? Do all species have similar proportions of living and dead cells in various plant tissue samples?

L345: why the focus on these methods? Is it because they also access different plant water domains than CVD?

L346: similar to other comments, is altering the right word choice to solely be used here? Are you not arguing that you are accessing different plant water pools too?

L357: first, "high transpiration moments"—Was the plant transpiring a lot after sunset? Second, why is isotopic difference between methods moreover a limitation? It seems like more of a justification to examine the possible information SPC extracted water would provide in many ecohydrologic studies.

L364: reads like this was a goal to have them comparable. similar to L298, seems at odds with introduction statement and possible advantage of SPC being more representative of transpired plant water than conventional CVD. Might be better to lead with "SPC accesses only part of the plant water fraction that CVD does and is therefore not an alternative to CVD in terms of plant water accessed." At the root of the issue is the word choice of alternative is unclear as mentioned in L298 comment.

Technical corrections:

L22: would it be better to state "longer" time rather than "long" time? Could be semantics, but seems like there is some ambiguity with what is meant by long time. One

month? One year? Multiple years?

L113: a prolongated or prolonged

L276: maybe "without extensive laboratory work" would be better here and similar places since each method inherently has different specific laboratory work

L329: "reach the leaves very rapidly" reads a little better.

References (not included in pre-print and referred to in this review):

Ellsworth, P.Z., Williams, D.G., 2007. Hydrogen isotope fractionation during water uptake by woody xerophytes. Plant Soil 291, 93–107. Morris, H., Plavcová, L., Cvecko, P., Fichtler, E., Gillingham, M.A.F., Martínez-Cabrera, H.I., Mcglinn, D.J., Wheeler, E., Zheng, J., Ziemińska, K., Jansen, S., 2016. A global analysis of parenchyma tissue fractions in secondary xylem of seed plants. New Phytol. 209, 1553–1565. Zhao, L., Wang, L., Cernusak, L.A., Liu, X., Xiao, H., Zhou, M., Zhang, S., 2016. Significant Difference in Hydrogen Isotope Composition Between Xylem and Tissue Water in Populus Euphratica. Plant Cell Environ. 39, 1848–1857.

---

## Referee Comment (RC2) · Anonymous Referee #2 · 18 Dec 2020

General comments:

The study conducted by Zuecco et al. addresses a very important issue in the ecohydrology community. They investigate two plant water extraction methods and attempt to analyse the differences of the isotopic composition of plant water in relation to the respectively used method. Namely, they compare plant water isotopic compositions extracted with either a Scholander pressure chamber (SPC) or cryogenic vacuum distillation (CVD). They aim at assessing (1) differences in isotopic composition related to the technique used (i.e. SPC or CVD) and additionally (2) want to analyse how differences in the isotopic composition are related to plant species or plant tissue type (from hereon, these two aims will be referred to using the terms aim1 and aim2). While this is a much needed study with a promising topic related to ecohydrology, I have, what I believe, are major concerns regarding the implementation of the method comparison.

While the authors used a fully suberized one- year old shoot to extract water from using the SPC method, they extracted water from several tissues making up a twig/shoot using CVD. These are not the same and therefore cannot be compared. Thus, I feel the authors cannot address aim1. Aim2 addresses differences in relation to tissue type and plant species, but only one sample type is used for SPC. Additionally, the extraction with SPC happened with attached leaves and unpeeled bark. The authors mention in the introduction and the discussion the Geißler et al. 2019 study, but do not take the same precautions when handling the samples (I know that this data set was taken prior to the Geißler manuscript, nevertheless the precautions are valid also for sample handling when not analysing water isotopes (e.g. when analysing xylem sap for nutrients the phloem is peeled off before the shoot is extracted with the SPC)). Is it not likely that when leaves are still attached and intact, to extract xylem sap from the leaves especially when applying high pressures (eg for the Ressi samples) and long extraction times (i.e. "until all the water was collected" L139/140)? Especially on the notice that the samples obtained with SPC in situ were different in colour, the authors could have taken another sample were they peeled the bark and stripped the leaves, thus assessing potential contamination of organics, and of the still living cells in attached leaves.

That said, I want to emphasize, that the topic and the idea for this study is needed, the community is waiting for this assessment and the results that can come from it, so I would encourage the authors to provide either better argumentation of why they can compare samples not from the same tissue, or an additional dataset, in which they compare the same tissues (e.g. complete shoot extracted with CVD or all plant tissues also extracted with SPC) and then address these aims again.

Additionally, I think the manuscript would benefit from a thorough review of a native speaker. Therefore, I will not comment on nor suggest improvements regarding language use or sentence structure.

Specific comments:

INTRODUCTION:

L 41: for completeness sake I would add Marshall et al. (2020) to the vapor equilibration method

L44: I would suggest giving either a comprehensive list of research in which this method is described/applied or one key publication in which the method is extensively described/first published

L45/46: "stored in dead and living cells for months or years" please add a reference

L49-54: Maybe shortening this section would benefit the reader, as they can look it up in the cited literature

L55ff: I would encourage you to shorten this section to the most relevant conclusion, i.e. that the results are ambiguous and different methods result in different outcomes

L69 adding Scholander (1966) to the references would help a comprehensive list (see reference list below)

L73: what about Ellsworth and Williams (2007) and Magh et al. (2020)? Even though Ellsworth and Williams did use SPC in a different way I think it would be worth mentioning with a statement like the one you make here.

STUDY SITE AND SAMPLING:

L88: Reference "Autonomous province of Bozen-Bolzano" is missing URL or in reference list. Please add

L89: what do you mean by "relatively mature"? please specify either the exact age

of the stands (for all three sites) or use appropriate describtion (i.e. either mature or juvenile)

FIG1: there are two red points on each photograph, I assume those are supposed to be the sampling sites? If so, one is really close to the river, the other one not. This probably has an influence on the isotopic composition of the samples and should be accounted for and definitely mentioned in the figure description.

L92+L104 Why after sunset? Especially considering water potentials were expected to be low due to the water deficit? Why not pre-dawn like at the Ressi site?

L105ff soil data are missing here but were added for the other sites. Please add here too.

MATERIALS AND METHODS:

L123 again missing the Scholander reference, please add

L137 please indicate if the samples were taken close to the river or not. The pressures used for extraction seem to indicate that there was no water deficit at the Laas site, but high deficit at the Ressi site, this could be related to the sampling position close to the river?

L140 how long did it take to sample all of the water for the samples? Were the times different for the different sites? Generally, are the samples all the same size and length?

L148ff The proclaimed target was to compare SPC to CVD. In order to do that it would have been good to extract the same tissues with both methods. Please elaborate why this has been done the way it was. Separating the twigs/shoots into smaller units to answer aim2 could still be done only for CVD if the small sample volume with SPC was the problem.

L153 what does this mean? The trees were too small? Too large? Please elaborate.

ISOTOPIC ANALYSIS:
L173-180 please add information about standardization, normalization, and possible corrections

DATA ANALYSIS:

L 190 I think this sentence could benefit from clarification. Maybe rephrase to something like this: "We report d-excess values enabling us to compare samples from different study areas."

L206ff This compares different plant tissues extracted by CVD with the whole shoot extracted using SPC. The results are not wrong, but misleading when considering the aim was to assess differences between the extraction methods. The tissues compared here are not the same and differences found from this analysis could also be related to differences in plant tissue instead of extraction method.

DISCUSSION:

L271 "in situ" sounds a little misleading because it is often used in relation to vapor equilibration methods. I suggest replacing it with "in the field".

L277 which advantages are these?

L280 extraction at 3Mpa likely damages living cells in the leaves/shoot, so it could be possible that the samples did not only consist of xylem sap anymore?

L283 how large are these volumes? Would it be possible to calculate a volume weighted isotopic composition of the samples CVD_L and CVD_TWB to get an approximation for the signature of a complete shoot extracted with CVD? Maybe that way, it would be possible to "compare" SPC and CVD with the same plant tissue used. That would only work if the extraction was complete though. Was that calculated? I mean were the CVD samples weighed before and after extraction and then again after drying?

L248ff I understand the volume from the SPC samples was too small for spectral contamination analysis, but an assessment for the CVD samples would have been possible? These data would be interesting especially when comparing the CVD_T and CVD_TWB samples.

L296-300 I feel this is redundant here because it was just discussed in the previous section. I suggest deleting

L310 I don't understand why it was expected for a CVD_WC sample obtained at breast height (?, this is an assumption, please specify the sampling heights and direction for all samples in the material and methods section) to have the same or a similar isotopic composition as a twig sampled from the crown (which likely has a different height). There has been evidence that this cannot be automatically assumed (de Deurwaerder et al. 2020). Please elaborate.

L313ff I don't think this argumentation is valid here. The samples are not comparable because they are not the same.

L319ff Again, this argument bases on the comparison between tissues which are not the same. I agree however, that SPC and CVD are able to extract different water pools/domains and it would be rational to quantify these differences especially when talking about water storage.

L326-329 redundant as the previous sentence says the same, rephrase or delete

L331 missing reference for water storage times. Please add

L339/340 how much older? Maybe a few examples would help here. Also, here you use plant water fractions before you used pools, please be consistent with which term you use throughout the manuscript.

General recommendations to improve the discussion:

I would suggest discussing the possibility of water pool mixture during transportation, because it is highly likely that water pools exchange and cannot be considered completely separate.

CONCLUDING REMARKS:

This reads like a second abstract/a summary and not like a conclusion. I suggest deleting the summarizing parts and keeping the main message (i.e. lines 363 -371)

References used in this review:

Deurwaerder HPT de, Visser MD, Detto M, Boeckx P, Meunier F, Kuehnhammer K, Magh RK, Marshall JD, Wang L, Zhao L, and Verbeeck H 2020. Causes and consequences of pronounced variation in the isotope composition of plant xylem water. Biogeosciences 17: 4853–4870.

Ellsworth PZ and Williams DG 2007. Hydrogen isotope fractionation during water uptake by woody xerophytes. Plant Soil 291: 93–107.

Magh R-K, Eiferle C, Burzlaff T, Dannenmann M, Rennenberg H, and Dubbert M 2020. Competition for water rather than facilitation in mixed beech-fir forests after drying-wetting cycle. J. Hydrol. 587: 124944.

Marshall JD, Cuntz M, Beyer M, Dubbert M, and Kuehnhammer K 2020. Borehole Equilibration: Testing a New Method to Monitor the Isotopic Composition of Tree Xylem Water in situ. Front. Plant Sci. 11: 1–14.

Scholander PF 1966. The role of solvent pressure in osmotic systems. P. Natl. Acad. Sci. 55: 1407–1414.

---

## Author Comment (AC1) · 27 Mar 2021

**Response to the reviewer 1**

**Manuscript: "A comparative study of plant water extraction methods for isotopic analyses: Scholander-type pressure chamber vs. cryogenic vacuum distillation", by Zuecco et al.**

We would like to thank the reviewer for the time (s)he spent on our manuscript and for the valuable suggestions. We report the reviewer's comments in black and their response below in blue.

This paper provides further examination of a promising plant water extraction method for isotopic analyses, Scholander-type pressure chamber water extraction (SPC), and compares results of that method to the conventional plant water extraction method of cryogenic vacuum distillation (CVD). This comparison is necessary to determine possible advantages, disadvantages, and implicit assumptions related to method choice for many ecohydrologic studies. However, I have what I think are valid concerns about the current version.

General feedback:

As mentioned in the introduction, there is an overarching issue within the ecohydrological community where differences in the isotopic composition of plant water have been related to extraction methods. These differences have been attributed to either one or a combination of the following: 1) Inter-lab differences: lab specific protocols, setups, accuracies, efficiencies 2) Inter-method differences: various alterations and various accessibility 2a) alterations of plant material associated with specific methods: e.g., fractionation effects associated with incomplete extractions or specific extraction methods, co-extraction of organic substances 2b) variability in the proportion of plant water domains accessed via different methods: lower residence time domain from xylem tracheary elements (dead cells) participating in transportation of water from soil to leaves versus higher residence time domain of living cells not participating in transportation of water. (I use domains rather than pools since that is the language recently being used for residence time of soil water in ecohydrology. Pools and domains are likely interchangeable and I am not saying one should be used over another, just that the use should be consistent throughout the paper. I will refer to them as domains for the remainder of this review to be consistent.)

For this paper the authors focus on (2a) with d-excess analyses, but there are issues (highlighted in specific comments) with whether d-excess can exclude considerations of (2b). Additionally, (2b) is poorly discussed for samples and analyses throughout paper. There are also some study design limitations for (2b) which would be hard to address by including more samples since sampling conditions will not be consistent, but the limitations should be mentioned for any future studies to address.

We agree with the reviewer that our study design has some limitations, and the manuscript lacks a specific discussion of the domains accessed by each method and sample type, and the possibility of extracting organic compounds by the two methods. In the revised version of the manuscript, we plan to significantly improve the Discussion and add a short section on the study limitations, presenting the gaps that our research could not fill in, and the way to overcome them in future studies. Despite the limitations highlighted by the reviewer, we still think that our study is important because, besides Geißler et al. (2019), previous research has never investigated the differences in the isotopic composition of plant water extracted by SPC and CVD and discussed the potentials and limitation of both methods.

The authors argue that the SPC method needs more investigation into its merits for ecohydrologic studies and that the SPC results need to be compared to conventional method of CVD. Hence, the authors two objectives (L79). However, there are issues with how they can address the first objective because their study design does not have a direct comparison between methods since the sampling materials are not the same.

Granted a shoot and a twig are similar, if not the same in some cases, but the use of these different terms is confusing from the onset.

We agree with the reviewer that the different terminology is a bit confusing. Therefore, we will use the term 'twig' for the SPC samples as well. The age of the twig can slightly differ between the SPC and CVD samples, but the diameters of CVD-T, CVD-TwB and SPC samples were comparable (3-6 mm), as well as the sampling position along the branch.

Furthermore, SPC samples are from one year old shoots with leaves and bark intact whereas CVD samples are from multiple tissue types that make up elements in a one year old shoot, but these elements are never all together (there are no CVD samples of one year old shoots with leaves and bark included).

We agree with the reviewer that this is, indeed, a limitation that will be discussed in the revised manuscript. However, we want to stress that we used, for SPC, the same sampling procedure widely applied by many ecophysiologists to measure the plant water potential.

For CVD, we are aware that this technique is able to extract the bulk plant water from different tissues and cells. Therefore, we decided not to make the comparison too complex by mixing twigs and leaves (by the way, it would have been difficult to collect the same amount/volume of twigs and leaves – and therefore the same plant water – for both SPC and CVD to have comparable samples). Furthermore, in the revised manuscript, we plan to discuss more in details that SPC samples are not comparable with a mix of water derived by CVD-TwB and CVD-L samples, particularly because SPC samples clearly did not show an evaporative signature (see the dual-isotope plots in Fig. 4).

These differences in tissue type for each method seem to be part of the reason for the second objective, but there is inadequate considerations of the (2b): variable access issues. With leaves still intact for SPC extraction, the possibility of extracting leaf water from predominantly xylem conducting cells in leaves seems hard to discount, especially when authors state that water extraction of shoots via SPC ended when they had collected all the water flowing out of the shoots (L139).

In the Introduction we reported that SPC is used to retrieve water in xylem conduits, and if we assume that there is no isotopic fractionation between xylem water present in the twig conduits and leaves conduits, then there should not be any difference in the isotopic composition of xylem water obtained from twigs or twigs and leaves together.

In addition, with bark still intact at collection site via SPC it is possible that phloem substrates are co-extracted, which is not mentioned when addressing how SPC samples were colorful in discussion(L285). In fact, I am perplexed when they cite Geißler et al. (2019) thoroughly in the introduction yet do not follow the same precautions, removing leaves and bark near collection site, to eliminate the contribution of water from leaves and phloem from live tissues still intact near collection site(L131).

Indeed, we did not follow the procedure described by Geißler et al. (2019), but we conducted the experiment before Geißler et al. (2019). Again, we want to stress that we carried out the SPC sampling procedure as it is usually done to measure the plant water potential.

We also agree with the reviewer that we cannot exclude the co-extraction of organic compounds by SPC, particularly in Ressi, where the water deficit conditions imposed us to apply higher pressure to extract the water samples compared to Laas/Lasa and Ahr/Aurino. We already mentioned this detail at lines 285-286. We will address this as a limitation of the SPC method and of our application.

In methods section 3.2, authors state that "Samples with and without bark were used to test whether the plant water extracted by the SPC method had an isotopic composition more similar to CVD-extracted bulk plant water (i.e., CVD-TwB) or to plant water deprived of phloem tissues (i.e., CVD-T)". This somewhat states the assumptions by the authors that a twigs with bark have more living cells, which is reasonable.

Furthermore, twigs without bark are predominantly dead cells and isotopic compositions of CVD-T samples should be more similar to isotopic composition of SPC samples (under the logic stated in introduction that SPC is collecting water primarily from dead cells). Although this logic holds up, this instance and other areas that discuss lignified tissues are missing the necessary consideration that total extractions of a twigs without bark via CVD could still be extracting water from living xylem parenchyma cells in the complex woody tissue. The proportion of xylem parenchyma varies by species and a recent global synthesis showed that the combined ray and axial parenchyma content for angiosperm trees and shrubs averaged 26.3 ± 12.4% (Morris et al., 2016). A possible contribution of a quarter of the water extracted seems like an important consideration to me, especially as they argue the possibility of the living cells having very different isotopic compositions than water being conducted in xylem conduits (L332). Granted some details of variation in dead cells are included in description of CVD-TcT (L153), but the relative amount of living cells needs to be highlighted more.

We thank the reviewer for providing the reference to Morris et al. (2016) that we will include in the revised manuscript. We also agree with the reviewer about the possibility of CVD to extract a significant amount of water from living parenchyma cells, which is an issue that we will address in more detail in the revised discussion and the new limitation section.

Overall, the authors need to discuss limitations in their design and provide arguments of why they can directly compare samples for first objective (e.g., SPC vs CVD-T). In addition, they need to have a more thorough discussion about the variability of accessing plant water domains with each method, tissue type, and species. In regard to methods, this includes many instances of when they compare to other studies investigating differences between methods and provide inadequate context to what plant water domains are accessed for the comparison. Finally, there needs to be greater clarity in the paper with the use of the word "effects" on isotopic composition of plant water. The paper lacks necessary discussion distinguishing the possible effects that could alter plant material versus effects of accessing different plant material when collecting a water sample, particularly when assessing results and conveying the future directions.

We agree that the current version of the Discussion should be revised by considering more the variability of accessing plant water domains by each method, tissue type and species, and by clarification of the effects on the isotopic composition of plant water. We plan to greatly revise the Discussion section by accounting for the indications of the reviewers, and by adding a section on limitations that will also address the directions for future experiments using SPC.

Specific comments:

L5: could benefit from more explicit mention of what plant water domain SPC is accessing similar to that done by CVD. In other words, why does SPC need more attention other than it being rarely applied?

We will rephrase the sentence highlighting that SPC could be used as a valid technique to extract xylem water that is likely transpiring during the sampling day.

L9: kind of misleading that SPC was done on multiple separate plant tissues when it was done from a one year old shoot comprised of multiple plant tissues. Certainly does not have to be here, but in the discussion it would be helpful to recognize past work on variability of isotopic composition of multiple plant materials done by Zhao et al. (2016) "Significant Difference in Hydrogen Isotope Composition Between Xylem and Tissue Water in Populus Euphratica"

We thank the reviewer for the suggested reference. We will rephrase the sentence to clarify that only CVD was performed on multiple plant tissues.

L18: granted you preface with likely, but how do you know that pressure applied via SPC is not affecting any living cells and only dead cells? Pressure is applied through leaves, so aren't some living cells impacted, albeit

minorly (especially considering your differences in CVD-L and SPC)? in your results you suggest cell walls were broken leading to co-extraction of organic substances by SPC.

We cannot exclude that we co-extracted organic compounds by SPC, particularly in Ressi where the water deficit conditions imposed us to apply higher pressure to extract the water samples compared to the Laas/Lasa and Ahr/Aurino sites. We already mentioned this detail at lines 285-286. We will address this as a limitation of the SPC method and of our application.

L35: mentioning these ecohydrologic studies is necessary, but your introduction currently lacks the broader context of issues within the community. CVD has been the conventional method for so long with assumptions that root water uptake is nonfractionating process for the most part and that total plant water via CVD has been considered representative of transpiration due to water in plant tissue assumed to be in equilibrium or well-mixed. The well-mixed assumption has been recently questioned by ecohydrologic separation studies (many of which you cite here) and discussion around plant water domains similar to soil water domains. For instance, you mention Barbeta et al. (2019) in the end of the discussion, but I feel like it would also help to include some of the ideas/issues from that paper here to help preface why SPC should be investigated more.

We thank the reviewer for this suggestion, and we will revise the Introduction by considering the findings by Barbeta et al. (2019).

L38: in addition to not altering plant material, these studies also want isotopic composition of plant water that is representative of transpiration which requires techniques that don't alter plant material and involves criticizing what plant water is accessed with each method (water from live versus dead cells).

We will reword this part.

L58: Millar et al. (2018) do categorize methods by what plant water domains(or pools in their case) are accessed and being more explicit about that here would help provide context to readers of why SPC is important to further investigate.

We will expand the sentences reporting the findings by Millar at al. (2018).

L61: what plant water domains were accessed with Fischer et al. (2019) various methods?

In the revised manuscript, we will expand the sentences reporting the findings by Fischer at al. (2019).

L71: what about the work by Ellsworth and Williams, 2007–"Hydrogen isotope fractionation during water uptake by woody xerophytes" which used modified SPC chamber to understand possible fractionation during root water uptake

We thank the reviewer for pointing this out; we will include the reference to Ellsworth and Williams (2007).

L75: This mentioned comparison by Geißler et al. (2019) does have important nuances of how they performed SPC method and the sample material preparation.

We agree with the reviewer, and we will expand the sentences reporting the findings by Geißler et al. (2019).

L93: what is the rationale in after sunset? Why not when water deficit is lower, i.e., pre-dawn to mid-morning, especially if there are issues in acquiring enough volume via SPC? "at the downstream in the Ahr/Aurino study area" is unclear – photo of site has two sites with one more downstream than the other which is also much closer to the stream.

We agree with the reviewer that the water deficit is even lower pre-dawn, but we were interested in carrying out the sampling when the transpiration fluxes were supposed to be close to their minimum (either during the early night or pre-dawn). Furthermore, due to logistic issues, in Laas/Lasa and Ahr/Aurino we were able to access the sampling sites only during the daylight and after the sunset.

We will revise Fig. 1 to make clear where the samples were collected.

Table 1: unclear what plant tissue/material was used for SPC, adding this detail to the table description would be helpful
We thank the reviewer for the indication, that we will integrate in the revised table.

L129: what position in canopy were one year old shoots taken from? Range in size of diameter extracted like that mentioned in 3.2 for CVD-T and CVD-TwB (L146)? Was canopy position and aspect similar for all sample types being more closely compared (SPC, CVD-T, CVD-TwB)?
Yes, the position of SPC, CVD-T and CVD-TwB samples was the same along the branch. The diameters of the twigs used for the samplings varied between 3 and 6 mm.

L131: what is the rationale in leaving the leaves on and bark intact near collection site? This does not follow rationale and guidance of Geißler et al. (2019) use of the SPC method to collect xylem water. Also, figure 2 has several leaves on the shoot in the chamber making the statement of "one or more leaves sealed inside chamber" somewhat unclear and added clarity would be beneficial.
As suggested by the reviewer, we will revise the sentence to improve the clarity. We are aware that we did not adopt same set up as Geißler et al. (2019), but we conducted the experiment before Geißler et al. (2019). Please, also refer to the replies that we have provided to the general comments.

L136: directly is a little vague, how does that differ from pipettes? Was the pressure chamber put on its side so that water could fall into vials directly via the help of gravity?
Yes, the SPC was put on its side to help the collection by gravity.

L139: rough timeframe? Longer or varied exposure to air/evaporation could impact comparison of results. All water implies that some water from leaves is incorporated in sample alongside shoot water which impacts how direct the comparisons can be between SPC and CVD-L/CVD-T/CVD-TwB
The sample collection by SPC was carried in less than 10 minutes for all the samples since the start of the pressure application. Furthermore, the plant water flowing out of the twigs was immediately trapped in the vial. Since the samplings were carried out during nighttime, we exclude a significant effect of evaporation on the isotopic composition of the samples. Indeed, average lc-excess were -1.0, -3.1 and 1.0 for SPC samples collected in Ressi, Ahr/Aurino and Laas/Lasa, respectively).

L148: what about living ray and axial parenchyma cells in xylem? similar or related reasoning for leaves is lacking in L145.
We will revise the sentence to improve the clarity about why we collected leaves samples as well for CVD extraction.

L150: was phloem removed for wood core samples? How much of active xylem/sapwood was used in wood core and did wood core include heartwood?
Yes, the phloem tissue was removed, and heartwood was not included. We will integrate this detail in the text.

L186: there are two additional LMWL sources missing here that are mentioned in figure 4 caption
We decided to depict the three LMWLs with the same color (pink) to improve the clarity of the figure and report the different equation for the three study sites in the caption.

L192: if you are trying to address whether method is fractionating water (or if you are accessing fractionated water) why not also do site specific lc-excess to more directly account for the local inputs? Especially with Ressi having very different LMWL this seems hard to gain much from the comparison with d-excess.

We decided to use d-excess for a direct comparison between the samples collected at the three study sites. As suggested by the reviewer, we will include a comparison based on lc-excess values in the revised manuscript.

L195: what do you mean by "effects"? Are the effects of the method altering isotopic composition? Or are they accessing different plant water domains? Or possibly both?

We thank the reviewer for pointing this out. In the revised manuscript, we will improve the clarity of the sentence.

L231: Not true for CVD-L samples. In general, CVD-L samples seem to be poorly integrated in discussion and results. I believe that their differences could help highlight that SPC with leaves attached are accessing different plant water domain in leaf than domains accessed by CVD for CVD-L.

We agree with the reviewer that the data of CVD-L samples were not well integrated in the results and the discussion. In the revised manuscript, we are providing a better inclusion of CVD-L samples in the text.

Figure 4: I think it would be helpful to include the GMWL on these plots to aid in visualizing how d-excess values are generated for each site.

We thank the reviewer for the suggestion, but we think that including the GMWL would affect the clarity of the plots. Furthermore, displaying the GMWL would not add much information to the findings that can be derived from the dual-isotope plots.

L277: does this include comparable in plant water domains extracted?

In this case, we were comparing the methods only in terms of simplicity and costs for plant water extraction. About the comparability in terms of plant water domains, we will expand and revise other parts of the Discussion.

L278: no specific mention of sampling times for each site/species sampled before this statement. Recommend putting SPC extraction times in methods

It will be done as suggested by the reviewer.

L286: what about phloem contributions via SPC extraction in addition to destruction of plant cell walls? According to methods the bark near the cut surface of one-year old shoot was not removed to limit contribution of phloem to extracted "water". What about contributions from water transporting and non-water transporting leaf cells? So, does your use of SPC method contradict opening statement of the discussion by breaking cell walls? Or does the contribution of phloem need greater consideration with your use of SPC method?

We will consider more the possible contribution of phloem as a limitation of the method and our application. We will discuss this limitation more in detail in the revised manuscript.

L298: based off introduction of CVD and SPC accessing different amount of plant water, is it really a "drawback" that they aren't comparable? I guess it is unclear if you are trying to state the SPC is an alternative for simplicity or if you are also considering SPC as an alternative for accessing water representative of transpiration vs total water via CVD?

We agree with the reviewer that this sentence is not very clear. We will revise the text reporting that SPC could be considered as an alternative for accessing water representative of transpiration.

L301: consider revising this sentence to be more clear, particularly the section starting in "indicating" is confusing as is. At a broader scope, being that leaves were still attached, then does this support that SPC is not accessing the evaporatively enriched pool of water in leaves that is found in the living leaf cells?
Yes, the very different signature of SPC samples compared to CVD-L supports the statement that SPC does not access the evaporated water domains that can be found in the living leaf cells.

L305: are you stating that the methods didn't fractionate plant water or that you didn't access fractionated water via both methods? Or both? I think this needs to be more clear for readers.
We meant both. Of course, this is a delicate point as we do not have unconfutable experimental evidence of the superimposition of the two issues, but we will revise this sentence to improve its clarity.

L306: I think it would be beneficial to point out that these values for Ressi are high with respect to other sites because the LMWL of Ressi plots above the GMWL in dual isotope space.
We will include this suggestion in the revised text.

L313: what plant water is sampled (predominantly transporting cells or nontransporting cells?) via direct vapor equilibration and microwave extraction and how does that help contextualize the comparisons of SPC results to CVD results? Although Zhao et al. (2018) attributed results to only fractionation effects rather than additionally considering access of different plant water domains, they also reported similar more negative d2H values for CVD-stem, CVD-core, CVD-root compared to d2H values of xylem sap sampled via needles.
In the revised manuscript, we will revise this part, while focusing more on the plant water domains accessed by the different methods.

L321: "sampling material" alone lacks distinction of different plant water domains and how the proportion of those domains possibly vary for each sampling material used
We will revise the sentence as suggested by the reviewer.

L332: Mask? Isn't the possibility of accessing different plant water domains a reason for comparison between methods and your objectives? It seems to me that the study design is what precludes you from a more direct comparison (sampling material not the exact same for both methods).
We will revise the sentence to include the possibility that SPC and CVD access different plant water domains. We do not think that our study design precludes the comparison between the two methods, as already stated in our first reply to the reviewer.

L342: any future directions on why there may be differences between species? Do all species have similar proportions of living and dead cells in various plant tissue samples?
In the new limitation section, we will provide more details about the future directions and other comparisons that could be made by using SPC.

L345: why the focus on these methods? Is it because they also access different plant water domains than CVD?
Since we and Geißler et al. (2019) have not compared SPC to other methods, besides CVD, we think that a wider comparison test is highly needed and useful to the ecohydrological community. This is the main novelty of our work that we will stress better in the revised version.

L346: similar to other comments, is altering the right word choice to solely be used here? Are you not arguing that you are accessing different plant water pools too?

We will revise the sentence to improve its clarity.

L357: first, "high transpiration moments" was the plant transpiring a lot after sunset? Second, why is isotopic difference between methods moreover a limitation? It seems like more of a justification to examine the possible information SPC extracted water would provide in many ecohydrologic studies.
We will remove "high transpiration moments" since our samplings were carried out during nighttime. Furthermore, we will rephrase the sentence to explain the access to different plant water domains.

L364: reads like this was a goal to have them comparable. similar to L298, seems at odds with introduction statement and possible advantage of SPC being more representative of transpired plant water than conventional CVD. Might be better to lead with "SPC accesses only part of the plant water fraction that CVD does and is therefore not an alternative to CVD in terms of plant water accessed." At the root of the issue is the word choice of alternative is unclear as mentioned in L298 comment.
We agree with the reviewer, and we will revise the sentence as suggested.

Technical corrections:
L22: would it be better to state "longer" time rather than "long" time? Could be semantics, but seems like there is some ambiguity with what is meant by long time. One month? One year? Multiple years?
We will use "long" rather than "longer".

L113: a prolongated or prolonged
We will replace "prolongated" with "prolonged".

L276: maybe "without extensive laboratory work" would be better here and similar places since each method inherently has different specific laboratory work
We agree with the revision.

L329: "reach the leaves very rapidly" reads a little better.
We agree with the revision.

References (not included in pre-print and referred to in this review):
Ellsworth, P.Z., Williams, D.G., 2007. Hydrogen isotope fractionation during water uptake by woody xerophytes. Plant Soil 291, 93–107.
Morris, H., Plavcová, L., Cvecko, P., Fichtler, E., Gillingham, M.A.F., Martínez-Cabrera, H.I., Mcglinn, D.J., Wheeler, E., Zheng, J., Ziemi´nska, K., Jansen, S., 2016. A global analysis of parenchyma tissue
fractions in secondary xylem of seed plants. New Phytol. 209, 1553–1565.
Zhao, L., Wang, L., Cernusak, L.A., Liu, X., Xiao, H., Zhou, M., Zhang, S., 2016. Significant Difference in Hydrogen Isotope Composition Between Xylem and Tissue Water in Populus Euphratica. Plant Cell Environ. 39, 1848–1857.

---

## Author Response (AR1)

Padova, Italy
31 August 2021

Dear Editor,

Please find enclosed our revised manuscript entitled "A comparative study of plant water extraction methods for isotopic analyses: Scholander-type pressure chamber vs. cryogenic vacuum distillation" by G. Zuecco et al., to be considered for publication in "Hydrology and Earth System Sciences", in the special issue on "Water, isotope and solute fluxes in the soil–plant–atmosphere interface: investigations from the canopy to the root zone".
We would thank the editor for handling our manuscript and the reviewers for their suggestions, which helped us to significantly improve the manuscript. We carefully considered all comments of the reviewers and the editor, and we revised the manuscript according to the submitted responses to the reviewers.

The major changes can be summarized as follows:
- we made a thorough revision of the introduction and the objectives, following the indications of the editor and the reviewers;
- we added more details in the description of the study sites and the samplings;
- we computed the line-conditioned excess (requested by reviewer 1) and the weighted mean isotopic composition (suggestion from reviewer 2), which determined changes to Table 1, Fig. 4 and Fig. 7;
- we made a thorough revision of the discussion by making major changes in section 5.2, and by adding the new section 5.3 about the limitations of our study;
- we revised the terminology, as requested by both reviewers (e.g., "plant water domains" instead of "plant water pools").

Please note that the manuscript with track changes contains the revised Figs. 2, 4, 7 and 8.

We confirm that this manuscript has not been published elsewhere and is not under consideration by another journal. All authors have approved the manuscript and agree with its re-submission.
Thank you for considering our manuscript.

Best regards,

Giulia Zuecco on behalf of all co-authors.

Please address all correspondence to:
Giulia Zuecco (giulia.zuecco@unipd.it), Department of Land, Environment, Agriculture and Forestry, University of Padova, via dell'Università 16, 35020 Legnaro (PD), Italy

**Author's response**

**Response to the reviewer 1**
**Manuscript: "A comparative study of plant water extraction methods for isotopic analyses:**
**Scholander-type pressure chamber vs. cryogenic vacuum distillation", by Zuecco et al.**

We would like to thank the reviewer for the time (s)he spent on our manuscript and for the valuable suggestions. We report the reviewer's comments in black and their response below in blue. Lines (L) refer to the revised manuscript with no track changes.

This paper provides further examination of a promising plant water extraction method for isotopic analyses, Scholander-type pressure chamber water extraction (SPC), and compares results of that method to the conventional plant water extraction method of cryogenic vacuum distillation (CVD). This comparison is necessary to determine possible advantages, disadvantages, and implicit assumptions related to method choice for many ecohydrologic studies. However, I have what I think are valid concerns about the current version.
General feedback:
As mentioned in the introduction, there is an overarching issue within the ecohydrological community where differences in the isotopic composition of plant water have been related to extraction methods. These differences have been attributed to either one or a combination of the following: 1) Inter-lab differences: lab specific protocols, setups, accuracies, efficiencies 2) Inter-method differences: various alterations and various accessibility 2a) alterations of plant material associated with specific methods: e.g., fractionation effects associated with incomplete extractions or specific extraction methods, co-extraction of organic substances 2b) variability in the proportion of plant water domains accessed via different methods: lower residence time domain from xylem tracheary elements (dead cells) participating in transportation of water from soil to leaves versus higher residence time domain of living cells not participating in transportation of water. (I use domains rather than pools since that is the language recently being used for residence time of soil water in ecohydrology. Pools and domains are likely interchangeable and I am not saying one should be used over another, just that the use should be consistent throughout the paper. I will refer to them as domains for the remainder of this review to be consistent.)
For this paper the authors focus on (2a) with d-excess analyses, but there are issues (highlighted in specific comments) with whether d-excess can exclude considerations of (2b). Additionally, (2b) is poorly discussed for samples and analyses throughout paper. There are also some study design limitations for (2b) which would be hard to address by including more samples since sampling conditions will not be consistent, but the limitations should be mentioned for any future studies to address.

We agree with the reviewer that our study design has some limitations, and the manuscript lacks a specific discussion of the domains accessed by each method and sample type, and the possibility of extracting organic compounds by the two methods. In the revised version of the manuscript, we significantly improved the discussion and added a new section 5.3 on the study limitations, presenting the gaps that our research could not fill in, and the way to overcome them in future studies. Despite the limitations highlighted by the reviewer, we still think that our study is important because, besides Geißler et al. (2019), previous research has never investigated the differences in the isotopic composition of plant water extracted by SPC and CVD and discussed the potentials and limitation of both methods.

The authors argue that the SPC method needs more investigation into its merits for ecohydrologic studies and that the SPC results need to be compared to conventional method of CVD. Hence, the

authors two objectives (L79). However, there are issues with how they can address the first objective because their study design does not have a direct comparison between methods since the sampling materials are not the same. Granted a shoot and a twig are similar, if not the same in some cases, but the use of these different terms is confusing from the onset.

We agree with the reviewer that the different terminology is a bit confusing. In the revised manuscript we used the term "twig" for the SPC samples as well. The age of the twig can slightly differ between the SPC and CVD samples, but the diameters of CVD-T, CVD-TwB and SPC samples were comparable (3-6 mm) (please see L160-161 and L182-183), as well as the sampling position along the branch (this detail was added in the description of the study sites).

Furthermore, SPC samples are from one year old shoots with leaves and bark intact whereas CVD samples are from multiple tissue types that make up elements in a one year old shoot, but these elements are never all together (there are no CVD samples of one year old shoots with leaves and bark included).

We agree with the reviewer that this is, indeed, a limitation discussed in the revised manuscript (L395-402 and L411-414). However, we want to stress that we used, for SPC, the same sampling procedure widely applied by many ecophysiologists to measure the plant water potential.

For CVD, we are aware that this technique is able to extract the total plant water from different tissues and cells. Therefore, we decided not to make the comparison too complex by mixing twigs and leaves (by the way, it would have been difficult to collect the same amount/volume of twigs and leaves – and therefore the same plant water – for both SPC and CVD to have comparable samples). Furthermore, in the revised manuscript, we discussed more in details that SPC samples are not comparable with a mixture of water derived by CVD-TwB and CVD-L samples, particularly because SPC samples clearly did not show an evaporative signature (see the dual-isotope plots in Fig. 4).

These differences in tissue type for each method seem to be part of the reason for the second objective, but there is inadequate considerations of the (2b): variable access issues. With leaves still intact for SPC extraction, the possibility of extracting leaf water from predominantly xylem conducting cells in leaves seems hard to discount, especially when authors state that water extraction of shoots via SPC ended when they had collected all the water flowing out of the shoots (L139).

In the introduction we reported that SPC is used to retrieve water in xylem conduits, and if we assume that there is no isotopic fractionation between xylem water present in the twig conduits and leaves conduits, then there should not be any difference in the isotopic composition of xylem water obtained from twigs or twigs and leaves together.

In addition, with bark still intact at collection site via SPC it is possible that phloem substrates are co-extracted, which is not mentioned when addressing how SPC samples were colorful in discussion(L285). In fact, I am perplexed when they cite Geißler et al. (2019) thoroughly in the introduction yet do not follow the same precautions, removing leaves and bark near collection site, to eliminate the contribution of water from leaves and phloem from live tissues still intact near collection site(L131).

Indeed, we did not follow the procedure described by Geißler et al. (2019), but we conducted the experiment before Geißler et al. (2019). Again, we want to stress that we carried out the SPC sampling procedure as it is usually done to measure the plant water potential.

We also agree with the reviewer that we cannot exclude the co-extraction of organic compounds by SPC, particularly in Ressi, where the water deficit conditions imposed us to apply higher pressure to extract the water samples compared to Laas/Lasa and Ahr/Aurino. We addressed this as a limitation of the SPC method and of our application (L403-410).

In methods section 3.2, authors state that "Samples with and without bark were used to test whether the plant water extracted by the SPC method had an isotopic composition more similar to CVD-extracted bulk plant water (i.e., CVD-TwB) or to plant water deprived of phloem tissues (i.e., CVD-T)". This somewhat states the assumptions by the authors that a twigs with bark have more living cells, which is reasonable. Furthermore, twigs without bark are predominantly dead cells and isotopic compositions of CVD-T samples should be more similar to isotopic composition of SPC samples (under the logic stated in introduction that SPC is collecting water primarily from dead cells). Although this logic holds up, this instance and other areas that discuss lignified tissues are missing the necessary consideration that total extractions of a twigs without bark via CVD could still be extracting water from living xylem parenchyma cells in the complex woody tissue. The proportion of xylem parenchyma varies by species and a recent global synthesis showed that the combined ray and axial parenchyma content for angiosperm trees and shrubs averaged 26.3 ± 12.4% (Morris et al., 2016). A possible contribution of a quarter of the water extracted seems like an important consideration to me, especially as they argue the possibility of the living cells having very different isotopic compositions than water being conducted in xylem conduits (L332). Granted some details of variation in dead cells are included in description of CVD-TcT (L153), but the relative amount of living cells needs to be highlighted more.

We thank the reviewer for providing the reference to Morris et al. (2016) that we included in the revised manuscript. We agree with the reviewer that CVD accesses various plant water domains, and this is an issue that we addressed in more detail in the revised discussion, and in comparison with SPC (L360-370).

Overall, the authors need to discuss limitations in their design and provide arguments of why they can directly compare samples for first objective (e.g., SPC vs CVD-T). In addition, they need to have a more thorough discussion about the variability of accessing plant water domains with each method, tissue type, and species. In regard to methods, this includes many instances of when they compare to other studies investigating differences between methods and provide inadequate context to what plant water domains are accessed for the comparison. Finally, there needs to be greater clarity in the paper with the use of the word "effects" on isotopic composition of plant water. The paper lacks necessary discussion distinguishing the possible effects that could alter plant material versus effects of accessing different plant material when collecting a water sample, particularly when assessing results and conveying the future directions.

We agree that the discussion should have considered more the variability of accessing plant water domains by each method, tissue type and species, and by clarification of the effects on the isotopic composition of plant water. We greatly revised section 5.2 by accounting for the indications of the reviewers, and by adding a section on limitations (5.3) that also addresses the directions for future experiments using SPC.

Specific comments:
L5: could benefit from more explicit mention of what plant water domain SPC is accessing similar to that done by CVD. In other words, why does SPC need more attention other than it being rarely applied?

We rephrased the sentence highlighting that SPC could be used as a valid technique to extract the more mobile plant water (L5-7).

L9: kind of misleading that SPC was done on multiple separate plant tissues when it was done from a one year old shoot comprised of multiple plant tissues. Certainly does not have to be here, but in the discussion it would be helpful to recognize past work on variability of isotopic composition of multiple

plant materials done by Zhao et al. (2016) "Significant Difference in Hydrogen Isotope Composition Between Xylem and Tissue Water in Populus Euphratica"

We thank the reviewer for the suggested reference. We rephrased the sentence to clarify that only CVD was performed on multiple plant tissues (L10).

L18: granted you preface with likely, but how do you know that pressure applied via SPC is not affecting any living cells and only dead cells? Pressure is applied through leaves, so aren't some living cells impacted, albeit minorly (especially considering your differences in CVD-L and SPC)? in your results you suggest cell walls were broken leading to co-extraction of organic substances by SPC.

We cannot exclude that we co-extracted organic compounds by SPC, particularly in Ressi where the water deficit conditions imposed us to apply higher pressure to extract the water samples compared to the Laas/Lasa and Ahr/Aurino sites. We addressed this as a limitation of the SPC method and of our application (L417-419).

L35: mentioning these ecohydrologic studies is necessary, but your introduction currently lacks the broader context of issues within the community. CVD has been the conventional method for so long with assumptions that root water uptake is nonfractionating process for the most part and that total plant water via CVD has been considered representative of transpiration due to water in plant tissue assumed to be in equilibrium or well-mixed. The well-mixed assumption has been recently questioned by ecohydrologic separation studies (many of which you cite here) and discussion around plant water domains similar to soil water domains. For instance, you mention Barbeta et al. (2019) in the end of the discussion, but I feel like it would also help to include some of the ideas/issues from that paper here to help preface why SPC should be investigated more.

We thank the reviewer for this suggestion. We revised the introduction by considering the findings by Barbeta et al. (2019) (L79-84).

L38: in addition to not altering plant material, these studies also want isotopic composition of plant water that is representative of transpiration which requires techniques that don't alter plant material and involves criticizing what plant water is accessed with each method (water from live versus dead cells).

We reworded this part (L39-40).

L58: Millar et al. (2018) do categorize methods by what plant water domains (or pools in their case) are accessed and being more explicit about that here would help provide context to readers of why SPC is important to further investigate.

We expanded the sentences reporting the findings by Millar at al. (2018) (L59-69).

L61: what plant water domains were accessed with Fischer et al. (2019) various methods?

In the revised manuscript, we expanded the sentences reporting the findings by Fischer at al. (2019) (L72-73).

L71: what about the work by Ellsworth and Williams, 2007—"Hydrogen isotope fractionation during water uptake by woody xerophytes" which used modified SPC chamber to understand possible fractionation during root water uptake

We thank the reviewer for pointing this out; we included the reference to Ellsworth and Williams (2007).

L75: This mentioned comparison by Geißler et al. (2019) does have important nuances of how they performed SPC method and the sample material preparation.

We agree with the reviewer, and we expanded the sentences reporting the findings by Geißler et al. (2019) (L93-97).

L93: what is the rationale in after sunset? Why not when water deficit is lower, i.e., pre-dawn to mid-morning, especially if there are issues in acquiring enough volume via SPC? "at the downstream in the Ahr/Aurino study area" is unclear – photo of site has two sites with one more downstream than the other which is also much closer to the stream.

We agree with the reviewer that the water deficit is even lower pre-dawn, but we were interested in carrying out the sampling when the transpiration fluxes were supposed to be close to their minimum (either during the early night or pre-dawn). Furthermore, due to logistic issues, in Laas/Lasa and Ahr/Aurino we were able to access the sampling sites only during the daylight and after the sunset (L118-120 and L132-134).

Table 1: unclear what plant tissue/material was used for SPC, adding this detail to the table description would be helpful

We thank the reviewer for the indication, that we integrated in the revised caption of Table 1.

L129: what position in canopy were one year old shoots taken from? Range in size of diameter extracted like that mentioned in 3.2 for CVD-T and CVD-TwB (L146)? Was canopy position and aspect similar for all sample types being more closely compared (SPC, CVD-T, CVD-TwB)?

Yes, the position of SPC, CVD-T and CVD-TwB samples was the same along the branch. The diameters of the twigs used for the samplings varied between 3 and 6 mm (see the reply to a previous comment and the details added to sections 2.1, 2.2, 2.3 and 3.1).

L131: what is the rationale in leaving the leaves on and bark intact near collection site? This does not follow rationale and guidance of Geißler et al. (2019) use of the SPC method to collect xylem water. Also, figure 2 has several leaves on the shoot in the chamber making the statement of "one or more leaves sealed inside chamber" somewhat unclear and added clarity would be beneficial.

We are aware that we did not adopt the same set up as Geißler et al. (2019), but we conducted the experiment before Geißler et al. (2019). Please, also refer to the replies that we have provided to the general comments.

L136: directly is a little vague, how does that differ from pipettes? Was the pressure chamber put on its side so that water could fall into vials directly via the help of gravity?

Yes, the SPC was put on its side to help the collection by gravity (please see the added detail at L168-169).

L139: rough timeframe? Longer or varied exposure to air/evaporation could impact comparison of results. All water implies that some water from leaves is incorporated in sample alongside shoot water which impacts how direct the comparisons can be between SPC and CVD-L/CVD-T/CVD-TwB

The sample collection by SPC was carried in less than 10 minutes for all the samples since the start of the pressure application. Furthermore, the plant water flowing out of the twigs was immediately trapped in the vial. Since the samplings were carried out during nighttime, we exclude a significant effect of evaporation on the isotopic composition of the samples, which is supported by lc-excess close to zero (Table 1 and Fig. 7).

L148: what about living ray and axial parenchyma cells in xylem? similar or related reasoning for leaves is lacking in L145.

We revised the sentence to improve the clarity about why we collected leaves samples as well for CVD extraction (L181-182).

L150: was phloem removed for wood core samples? How much of active xylem/sapwood was used in wood core and did wood core include heartwood?
Yes, the phloem tissue was removed, and heartwood was not included. We integrate this detail at L188-189).

L186: there are two additional LMWL sources missing here that are mentioned in figure 4 caption
We decided to depict the three LMWLs with the same color (pink) to improve the clarity of the figure. In the revised Fig. 4, we added the equations of the three LMWLs inside the plots.

L192: if you are trying to address whether method is fractionating water (or if you are accessing fractionated water) why not also do site specific lc-excess to more directly account for the local inputs? Especially with Ressi having very different LMWL this seems hard to gain much from the comparison with d-excess.
As suggested by the reviewer, we included a comparison based on lc-excess values in the revised manuscript (please see the revised Table 1 and Fig. 7, and L292-299).

L195: what do you mean by "effects"? Are the effects of the method altering isotopic composition? Or are they accessing different plant water domains? Or possibly both?
In the revised manuscript, we changed the sentence to improve its clarity (L241-244).

L231: Not true for CVD-L samples. In general, CVD-L samples seem to be poorly integrated in discussion and results. I believe that their differences could help highlight that SPC with leaves attached are accessing different plant water domain in leaf than domains accessed by CVD for CVD-L.
We agree with the reviewer that the data of CVD-L samples were not well integrated in the results and the discussion. In the revised manuscript, we provided a better inclusion of CVD-L samples in the text.

Figure 4: I think it would be helpful to include the GMWL on these plots to aid in visualizing how d-excess values are generated for each site.
We thank the reviewer for the suggestion, but we think that including the GMWL would affect the clarity of the plots. Furthermore, displaying the GMWL would not add much information to the findings that can be derived from the dual-isotope plots.

L277: does this include comparable in plant water domains extracted?
In this case, we were comparing the methods only in terms of simplicity and costs for plant water extraction. About the comparability in terms of plant water domains, we expanded and greatly revised section 5.2.

L278: no specific mention of sampling times for each site/species sampled before this statement. Recommend putting SPC extraction times in methods
This detail was added at L175.

L286: what about phloem contributions via SPC extraction in addition to destruction of plant cell walls? According to methods the bark near the cut surface of one-year old shoot was not removed to limit contribution of phloem to extracted "water". What about contributions from water transporting and non-water transporting leaf cells? So, does your use of SPC method contradict opening statement of

the discussion by breaking cell walls? Or does the contribution of phloem need greater consideration with your use of SPC method?

We considered more the possible contribution of phloem as a limitation of the method and our application (L395-402).

L298: based off introduction of CVD and SPC accessing different amount of plant water, is it really a "drawback" that they aren't comparable? I guess it is unclear if you are trying to state the SPC is an alternative for simplicity or if you are also considering SPC as an alternative for accessing water representative of transpiration vs total water via CVD?

We agree with the reviewer that this sentence was not very clear. We considered the comments of reviewer 2 as well, and therefore, we decided to remove the sentence.

L301: consider revising this sentence to be more clear, particularly the section starting in "indicating" is confusing as is. At a broader scope, being that leaves were still attached, then does this support that SPC is not accessing the evaporatively enriched pool of water in leaves that is found in the living leaf cells?

Yes, the very different signature of SPC samples compared to CVD-L supports the statement that SPC does not access the evaporated water domains that can be found in the living leaf cells.

L305: are you stating that the methods didn't fractionate plant water or that you didn't access fractionated water via both methods? Or both? I think this needs to be more clear for readers.

We meant both. Of course, this is a delicate point as we do not have unconfutable experimental evidence of the superimposition of the two issues. We revised this sentence to improve its clarity (L348-349).

L306: I think it would be beneficial to point out that these values for Ressi are high with respect to other sites because the LMWL of Ressi plots above the GMWL in dual isotope space.

We modified the sentence, by referring to lc-excess values (L345-348).

L313: what plant water is sampled (predominantly transporting cells or nontransporting cells?) via direct vapor equilibration and microwave extraction and how does that help contextualize the comparisons of SPC results to CVD results? Although Zhao et al. (2018) attributed results to only fractionation effects rather than additionally considering access of different plant water domains, they also reported similar more negative d2H values for CVD-stem, CVD-core, CVD-root compared to d2H values of xylem sap sampled via needles.

In the revised manuscript, we deleted the sentence.

L321: "sampling material" alone lacks distinction of different plant water domains and how the proportion of those domains possibly vary for each sampling material used.

We revised the sentence as suggested by the reviewer (L360-363).

L332: Mask? Isn't the possibility of accessing different plant water domains a reason for comparison between methods and your objectives? It seems to me that the study design is what precludes you from a more direct comparison (sampling material not the exact same for both methods).

We removed the term "mask" and revised the sentence to improve its clarity (L370-372). We do not think that our study design precludes the comparison between the two methods, as already stated in our first reply to the reviewer.

L342: any future directions on why there may be differences between species? Do all species have similar proportions of living and dead cells in various plant tissue samples?
In the new limitation section 5.3, we provide more details about the future directions and other comparisons that could be made by using SPC.

L345: why the focus on these methods? Is it because they also access different plant water domains than CVD?
Since we and Geißler et al. (2019) have not compared SPC to other methods, besides CVD, we think that a wider comparison test is highly needed and useful to the ecohydrological community. This is the main novelty of our work that we stressed better in the revised version.

L346: similar to other comments, is altering the right word choice to solely be used here? Are you not arguing that you are accessing different plant water pools too?
We think that the term "altering" is the right word choice, but we also added the reference to the different plant water domains accessed by each method (L386-389).

L357: first, "high transpiration moments" was the plant transpiring a lot after sunset? Second, why is isotopic difference between methods moreover a limitation? It seems like more of a justification to examine the possible information SPC extracted water would provide in many ecohydrologic studies.
We removed "high transpiration moments" since our samplings were carried out during nighttime.

L364: reads like this was a goal to have them comparable. similar to L298, seems at odds with introduction statement and possible advantage of SPC being more representative of transpired plant water than conventional CVD. Might be better to lead with "SPC accesses only part of the plant water fraction that CVD does and is therefore not an alternative to CVD in terms of plant water accessed." At the root of the issue is the word choice of alternative is unclear as mentioned in L298 comment.
We agree with the reviewer, and we revised the sentence as suggested (L439-440).

Technical corrections:
L22: would it be better to state "longer" time rather than "long" time? Could be semantics, but seems like there is some ambiguity with what is meant by long time. One month? One year? Multiple years?
As suggested by the reviewer, we used "longer" than "long".

L113: a prolongated or prolonged
We replaced "prolongated" with "prolonged".

L276: maybe "without extensive laboratory work" would be better here and similar places since each method inherently has different specific laboratory work
We agree with the revision.

L329: "reach the leaves very rapidly" reads a little better.
We agree with the revision.

References (not included in pre-print and referred to in this review):
Ellsworth, P.Z., Williams, D.G., 2007. Hydrogen isotope fractionation during water uptake by woody xerophytes. Plant Soil 291, 93–107.
Morris, H., Plavcová, L., Cvecko, P., Fichtler, E., Gillingham, M.A.F., Martínez-Cabrera, H.I., Mcglinn, D.J., Wheeler, E., Zheng, J., Ziemi´nska, K., Jansen, S., 2016. A global analysis of parenchyma tissue fractions in secondary xylem of seed plants. New Phytol. 209, 1553–1565.

Zhao, L., Wang, L., Cernusak, L.A., Liu, X., Xiao, H., Zhou, M., Zhang, S., 2016. Significant Difference in Hydrogen Isotope Composition Between Xylem and Tissue Water in Populus Euphratica. Plant Cell Environ. 39, 1848–1857.

**Response to the reviewer 2**
**Manuscript: "A comparative study of plant water extraction methods for isotopic analyses: Scholander-type pressure chamber vs. cryogenic vacuum distillation", by Zuecco et al.**

We would like to thank the reviewer for the time (s)he spent on our manuscript and for the valuable suggestions. We report the reviewer's comments in black and their response below in blue.

General comments:
The study conducted by Zuecco et al. addresses a very important issue in the ecohydrology community. They investigate two plant water extraction methods and attempt to analyse the differences of the isotopic composition of plant water in relation to the respectively used method. Namely, they compare plant water isotopic compositions extracted with either a Scholander pressure chamber (SPC) or cryogenic vacuum distillation (CVD). They aim at assessing (1) differences in isotopic composition related to the technique used (i.e. SPC or CVD) and additionally (2) want to analyse how differences in the isotopic composition are related to plant species or plant tissue type (from hereon, these two aims will be referred to using the terms aim1 and aim2). While this is a much needed study with a promising topic related to ecohydrology, I have, what I
believe, are major concerns regarding the implementation of the method comparison.
While the authors used a fully suberized one- year old shoot to extract water from using the SPC method, they extracted water from several tissues making up a twig/shoot using CVD. These are not the same and therefore cannot be compared. Thus, I feel the authors cannot address aim1. Aim2 addresses differences in relation to tissue type and plant species, but only one sample type is used for SPC. Additionally, the extraction with SPC happened with attached leaves and unpeeled bark. The authors mention in the introduction and the discussion the Geißler et al. 2019 study, but do not take the same precautions when handling the samples (I know that this data set was taken prior to the Geißler manuscript, nevertheless the precautions are valid also for sample handling when not analysing water isotopes (e.g. when analysing xylem sap for nutrients the phloem is peeled off before the shoot is extracted with the SPC)).

We agree with the reviewer that our study design has some limitations, but we still think that our study is very important for the ecohydrological community because, besides Geißler et al. (2019), previous research has never investigated the differences in the isotopic composition of plant water extracted by SPC and CVD and discussed the potentials and limitation of both methods. We are aware that in various studies focusing on nutrients, the bark is peeled off, but in our case, we still considered various options for CVD tissues, one of which comprises the presence of bark. Furthermore, we still think that the twig samples used for CVD-T and CVD-TwB and the ones used for SPC were very similar because they had very similar size (range between 3 and 6 mm), were lignified and were collected at the same position in the tree.

Is it not likely that when leaves are still attached and intact, to extract xylem sap from the leaves especially when applying high pressures (eg for the Ressi samples) and long extraction times (i.e. "until all the water was collected" L139/140)? Especially on the notice that the samples obtained with SPC in situ were different in colour, the authors could have taken another sample were they peeled the bark and stripped the leaves, thus assessing potential contamination of organics, and of the still living cells in attached leaves.

We exclude that there was an influence of leaf water on the isotopic composition of SPC samples because CVD-L samples clearly had an evaporative signature (see the dual-isotope plots in Fig. 4) compared to SPC samples. Furthermore, in the introduction we reported that SPC is used to retrieve water in xylem conduits, and if we assume that there is no isotopic fractionation between xylem water present in the twig conduits and leaves conduits, then there should not be any difference in the isotopic composition of xylem water obtained from twigs or twigs and leaves together.

That said, I want to emphasize, that the topic and the idea for this study is needed, the community is waiting for this assessment and the results that can come from it, so I would encourage the authors to provide either better argumentation of why they can compare samples not from the same tissue, or an additional dataset, in which they compare the same tissues (e.g. complete shoot extracted with CVD or all plant tissues also extracted with SPC) and then address these aims again.
We thank the reviewer for appreciating our work. We hope that the revised manuscript clarifies all the questions of the two reviewers.

Additionally, I think the manuscript would benefit from a thorough review of a native speaker. Therefore, I will not comment on nor suggest improvements regarding language use or sentence structure.
We thank the reviewer for the suggestion. The manuscript was carefully checked for grammar and spelling errors.

Specific comments:
INTRODUCTION:
L 41: for completeness sake I would add Marshall et al. (2020) to the vapor equilibration method
Done.

L44: I would suggest giving either a comprehensive list of research in which this method is described/applied or one key publication in which the method is extensively described/first published
As requested, in the revised manuscript, we added some key references about CVD (L44).

L45/46: "stored in dead and living cells for months or years" please add a reference
We modified the sentence, and added a reference (L49-50).

L49-54: Maybe shortening this section would benefit the reader, as they can look it up in the cited literature
We decided to keep the sentences, but we made some minor changes (L54-58).

L55ff: I would encourage you to shorten this section to the most relevant conclusion, i.e. that the results are ambiguous and different methods result in different outcomes
We thank the reviewer for the suggestion. However, in contrast to this comment, the first reviewer asked to expand this section, by adding more details about the findings by Millar et al. (2018) and Fischer et al. (2019). We tried to revise the text by keeping only the key results (L59-76).

L69 adding Scholander (1966) to the references would help a comprehensive list (see reference list below)
We thank the reviewer for the suggested reference.

L73: what about Ellsworth and Williams (2007) and Magh et al. (2020)? Even though Ellsworth and Williams did use SPC in a different way I think it would be worth mentioning with a statement like the one you make here.

We thank the reviewer for the suggested references, that we added in the revised manuscript.

STUDY SITE AND SAMPLING:
L88: Reference "Autonomous province of Bozen-Bolzano" is missing URL or in reference list. Please add

We modified the reference.

L89: what do you mean by "relatively mature"? please specify either the exact age of the stands (for all three sites) or use appropriate description (i.e. either mature or juvenile)

We thank the reviewer for pointing this out. We modified the sentence to clarify what we meant (L114-115).

FIG1: there are two red points on each photograph, I assume those are supposed to be the sampling sites? If so, one is really close to the river, the other not. This probably has an influence on the isotopic composition of the samples and should be accounted for and definitely mentioned in the figure description.

In sections 2.1, 2.2 and 2.3, we clarified where the samples were collected.

L92+L104 Why after sunset? Especially considering water potentials were expected to be low due to the water deficit? Why not pre-dawn like at the Ressi site?

We agree with the reviewer that the water deficit is even lower pre-dawn, but we were interested in carry out the sampling when the transpiration fluxes were supposed to be close to their minimum (either during the early night or pre-dawn). Furthermore, due to logistic issues, in Laas/Lasa and Ahr/Aurino we were able to access the sampling sites only during the daylight and after the sunset.

L105ff soil data are missing here but were added for the other sites. Please add here too.

We added the soil texture data.

MATERIALS AND METHODS:
L123 again missing the Scholander reference, please add

Done.

L137 please indicate if the samples were taken close to the river or not. The pressures used for extraction seem to indicate that there was no water deficit at the Laas site, but high deficit at the Ressi site, this could be related to the sampling position close to the river?

The orchards in Laas/Lasa are generally irrigated during dry periods. This irrigation can explain the lower water deficit in Laas/Lasa compared to Ressi, which is mainly covered by a deciduous forest.

L140 how long did it take to sample all of the water for the samples? Were the times different for the different sites? Generally, are the samples all the same size and length?

The duration of the extraction was different among the samples (probably due to the different water deficit conditions), but we kept it as short as possible (less than 10 minutes) to minimize the evaporation. We tried to use samples of similar size for all the species and the three sites.

L148ff The proclaimed target was to compare SPC to CVD. In order to do that it would have been good to extract the same tissues with both methods. Please elaborate why this has been done the way it

was. Separating the twigs/shoots into smaller units to answer aim2 could still be done only for CVD if the small sample volume with SPC was the problem.

We still think that the twig samples used for CVD-T and CVD-TwB and the ones used for SPC were very similar because they had very similar size (range between 3 and 6 mm), were lignified and were collected at the same position in the tree. However, as already mentioned by both reviewers, for SPC we cannot exclude a contamination by other tissues, such as phloem. Therefore, we added a new section 5.3 presenting the limitations of our study, and we expanded the discussion at section 5.2 about the capability of the two extraction methods to access different plant water domains.

L153 what does this mean? The trees were too small? Too large? Please elaborate.

Yes, these trees had small diameters and they are located in private land. Therefore, we chose not to use increment borers to avoid additional damages to the trees which were also used for sapflow measurements and during further sampling campaigns. We added more details in the revised manuscript at L189-191.

ISOTOPIC ANALYSIS:

L173-180 please add information about standardization, normalization, and possible corrections

We added more details about the isotopic analyses at L225-226.

DATA ANALYSIS:

L 190 I think this sentence could benefit from clarification. Maybe rephrase to something like this: "We report d-excess values enabling us to compare samples from different study areas."

We thank the reviewer for the suggestion. However, based on the comments of reviewer 1, we now refer to lc-excess.

L206ff This compares different plant tissues extracted by CVD with the whole shoot extracted using SPC. The results are not wrong, but misleading when considering the aim was to assess differences between the extraction methods. The tissues compared here are not the same and differences found from this analysis could also be related to differences in plant tissue instead of extraction method.

Please see the reply given to the comment for L148ff. In the new limitation section, we thoroughly discussed the capability of the two extraction methods to access different plant water domains, and how to improve our sampling design in future experiments.

DISCUSSION:

L271 "in situ" sounds a little misleading because it is often used in relation to vapor equilibration methods. I suggest replacing it with "in the field".

Done.

L277 which advantages are these?

We slightly modified the sentence to improve its clarity (L328-329).

L280 extraction at 3Mpa likely damages living cells in the leaves/shoot, so it could be possible that the samples did not only consist of xylem sap anymore?

Yes, it is possible that such samples could not include only xylem water, but also water extracted from living cells. We added this part as a limitation of our study.

L283 how large are these volumes? Would it be possible to calculate a volume weighted isotopic composition of the samples CVD_L and CVD_TWB to get an approximation for the signature of a

complete shoot extracted with CVD? Maybe that way, it would be possible to "compare" SPC and CVD with the same plant tissue used.

That would only work if the extraction was complete though. Was that calculated? I mean were the CVD samples weighed before and after extraction and then again after drying?

We thank the reviewer for the suggestion. Yes, the samples were weighed before and after the extraction to determine the extraction efficiency for CVD. We observed an average extraction efficiency for the specific CVD system of 98.6% (n=65), whereas the median was 100%. In the revised manuscript, we provide the computation of this volume-weighted isotopic composition in Table 1 and Fig. 7.

L248ff I understand the volume from the SPC samples was too small for spectral contamination analysis, but an assessment for the CVD samples would have been possible? These data would be interesting especially when comparing the CVD_T and CVD_TWB samples.

We agree with the reviewer that it would have been interesting to have spectral contamination data. However, it was not measured, and it will not be possible to perform the analysis now.

L296-300 I feel this is redundant here because it was just discussed in the previous section. I suggest deleting

We agree with the reviewer, and we removed these lines.

L310 I don't understand why it was expected for a CVD_WC sample obtained at breast height (?, this is an assumption, please specify the sampling heights and direction for all samples in the material and methods section) to have the same or a similar isotopic composition as a twig sampled from the crown (which likely has a different height). There has been evidence that this cannot be automatically assumed (de Deurwaerder et al. 2020). Please elaborate.

We are aware and we have not automatically assumed that the various samples had the same isotopic composition. Indeed, we decided to test whether they had the same or similar isotopic signature.

L313ff I don't think this argumentation is valid here. The samples are not comparable because they are not the same.

Please see the reply given to the comment for L148ff and other previous comments.

L319ff Again, this argument bases on the comparison between tissues which are not the same. I agree however, that SPC and CVD are able to extract different water pools/domains and it would be rational to quantify these differences especially when talking about water storage.

We agree with the reviewer, and we expanded the discussion about the capability of the two extraction methods to access different plant water domains.

L326-329 redundant as the previous sentence says the same, rephrase or delete

We thank the reviewer for the suggestion. We removed the sentence.

L331 missing reference for water storage times. Please add

We added a reference about water storage times.

L339/340 how much older? Maybe a few examples would help here. Also, here you use plant water fractions before you used pools, please be consistent with which term you use throughout the manuscript.

As suggested by the first reviewer, we were more consistent in the terminology throughout the manuscript, by referring to "plant water domains".

General recommendations to improve the discussion:
I would suggest discussing the possibility of water pool mixture during transportation, because it is highly likely that water pools exchange and cannot be considered completely separate.
We agree with the reviewer, and we considered the possibility of water pool mixture transportation in the revised discussion (L379-381).

CONCLUDING REMARKS:
This reads like a second abstract/a summary and not like a conclusion. I suggest deleting the summarizing parts and keeping the main message (i.e. lines 363 -371)
We thank the reviewer for the suggestion. We tried to shorten a bit the concluding remarks, but we also considered the suggestions of reviewer 1.

References used in this review:
Deurwaerder HPT de, Visser MD, Detto M, Boeckx P, Meunier F, Kuehnhammer K, Magh RK, Marshall JD, Wang L, Zhao L, and Verbeeck H 2020. Causes and consequences of pronounced variation in the isotope composition of plant xylem water. Biogeosciences 17: 4853–4870.
Ellsworth PZ and Williams DG 2007. Hydrogen isotope fractionation during water uptake by woody xerophytes. Plant Soil 291: 93–107.
Magh R-K, Eiferle C, Burzlaff T, Dannenmann M, Rennenberg H, and Dubbert M 2020. Competition for water rather than facilitation in mixed beech-fir forests after dryingwetting cycle. J. Hydrol. 587: 124944.
Marshall JD, Cuntz M, Beyer M, Dubbert M, and Kuehnhammer K 2020. Borehole Equilibration: Testing a New Method to Monitor the Isotopic Composition of Tree Xylem Water in situ. Front. Plant Sci. 11: 1–14.
Scholander PF 1966. The role of solvent pressure in osmotic systems. P. Natl. Acad. Sci. 55: 1407–1414.

---

## Referee Report (RR1)

**Much improved version of the manuscript, thanks.**

**Abstract**. No comments

**Introduction**:
I think you greatly improved the introduction and I have only some minor comments:

Line 39-40: water representative of transpiration needed in ecohydrological studies. With your findings the method of choice would be SPC. Pick that thread up in the discussion again.

Line 80-81: Barbeta et al. observed this depletion , but the problem that it was related to CVD and not the plant fractionation was identified by Chen et al. 2020, and Allen and Kirchner 2021). If you could add these references (see below) and put them in context with Barbeta et al. 2019 that would be great.

**Site descriptions**:

Line 109 and 113 what do you want to tell the reader with the Engel et al. in review reference? To look for a better site description in there? I think it is irrelevant and I would delete it, especially because this has not been published yet.

Line 125 again, why is it important to have this reference here? I think your site description is sufficient as is, but if you insist on adding a reference here, please add what it is for.

Table 1 belongs in the Results section, but while looking at it, how do you deal with the tiny deviations from the LMWL (i.e. the small lc values)? See (Landwehr and Coplen 2006) who indicate that lc's smaller than can be detected by the measurement precision are considered not to differ from LMWL (here , it looks like this could be the case for Alder CVD-WC, Chestnut CVD-T, Beech SPC and CVD-TwB and -TcT and maybe even more) could you add the calculations for S in regard to the measurement precision of the used instruments (as you write in the MM 2.5permil for delta2H and 0.1permil for delta18O) and then indicate in parentheses which values are and are not different from the LMWL in your case? It would give the reader the chance to directly assess differences.

Line 136 and again, emphasize what you want to say with adding these references here, or delete them if they are not giving additional information.

**Material and Methods**:

Line 234 the landwehr and Coplen reference is dated to 2004 in your manuscript, however when I checked it again on google scholar and web of science the date is 2006, please change it to the correct date.

Generally, I am missing the explanation of how and why you weighted the means of the measurements? I know it says it in the header of table 1 but it should be part of the MM for completeness sake. What did you define as the mean extracted volume? Did that then differ for each method or each species or each tissue? And if so, what is the benefit of weighting them, the measurement device takes the same amount of sample for injection, right?

Also, I think I was misunderstood in the last revision round when I asked for a volume weighted isotopic composition, I meant if you could do a mixing ratio calculation where (in theory) you would mix the samples CVD_L and CVD_TWB together (weighted by each their volume) and get a mixed isotope signal that would in theory then represent the same tissue as when you extracted one twig with bark and leaves using the SPC. But since you do not see any co-extraction of water from the leaves in the SPC samples, I can agree how you would most closely compare them to CVD-TwB. However, that does not mean they are the same tissue type.

**Results**:

Figure 4: If possible, it would be great if you could change the pink line and text to e.g. grey but that's just a personal preference, I feel the pink is too much, maybe other readers think so too?
Also, it would be reader friendly if you could plot the leaf data as an inset or a separate column and therewith zoom in to the data plotting closer to the LMWL. Especially looking at the Laas data, I think the figure would benefit greatly from a higher resolution on the potentially non-enriched data.

Line 297: I think once you correct for the measurement precision this will not be distinctly different from 0, so I think it is important that you add this S (as standard deviation) according to Landwehr and Coplen 2006.

**Discussion**:

Figure 8 should go to the last page of the result section.

I would switch sections 5.1 and 5.2 as it is logical to first read your assessment about the differences and therewith the answer to your main question: do these methods yield different results, and then move on with the method of choice discussion in relevance to ecohydrological questions.

Line 333 -336: This could also be related to the contamination by phloem, as you did not peel it from the twig. Please add this information here.

Line 347: again, I think if you would "correct" the lc-excess values by the precision of your measurement device, these values will be indistinguishable from 0 and therewith indistinguishable from the LMWL.

Lines 351-352: please discuss this also in relation to (Chen et al. 2020, Allen and Kirchner 2021)

**Concluding remarks**: no comments

References:

Allen S and Kirchner J 2021. Potential effects of cryogenic extraction biases on inferences drawn from xylem water deuterium isotope ratios: case studies using stable isotopes to infer plant water sources. Hydrol. Earth Syst. Sci. Discuss.: 1–15.

Chen Y, Helliker BR, Tang X, Li F, Zhou Y, and Song X 2020. Stem water cryogenic extraction biases estimation in deuterium isotope composition of plant source water. Proc. Natl. Acad. Sci. 117: 33345–33350.

Landwehr JM and Coplen TB 2006. Line-conditioned excess: a new method for characterizing stable hydrogen and oxygen isotope ratios in hydrologic systems. Int. Conf. Isot. Environ. Stud.: 132–135.

---

## Referee Report (RR2)

Review 3 Hess-2020-446

**A comparative study of plant water extraction methods for isotopic analyses: Scholander-type pressure chamber vs. cryogenic vacuum distillation**

General comments:

I will dive right in, since I summarized the manuscript the last two times, I reviewed it.

I think I have to apologize for not making my points clear enough, I will try to here:

In the results section of this revised version of the manuscript the authors compare every CVD extracted tissue to SPC extracted tissue.
The only comparison I think the authors can argue to be valid is that of SPC and CVDTwB and therefore, in my opinion, the only comparison that should be carried out and communicated in the results section.
It is an **a priori** assumption that the methods are **not** comparable for all the other plant tissues you extracted with CVD and therefore it is not an issue to be discussed but a comparison that should not be carried out in the first place.

Therefore my suggestion would be for you to take this under consideration and revise the manuscript accordingly. Especially in these sections 3.4 Data analysis, 4 Results, and 5. Discussion you should refrain from comparing all but CVDTwB and SPC.

I'm very sorry if this is my fault for not communicating this strongly enough.

I think you can very well communicate the other CVD extracted data as a comparison of plant tissue differences but not to compare the two methods.

I would also again encourage you to have a native speaker check the revised version.

**Specific comments:**

Introduction:
L48 the extraction of water directly from the stem due to positive pressure on the inside is only possible for very few tree species, otherwise no one would have to ever extract any plant tissues. Please make sure to clarify this when adding that reference (Zhao 2016)

L64 which two versions of CVD are you referring to here?

L94 check formatting of references

L114ff Here you switch to present tense while the rest of the introduction is written in past tense. I would suggest choosing either one consistently throughout the manuscript.

Material and Methods

L188 The bark and the leaves remain attached to the twig on what basis? In physiology the bark is usually removed to avoid phloem contamination. If you insist on keeping this reference here, I would ask you to dig a little bit into physiology research literature and at least mention that this is not the standard procedure when extracting xylem using the Scholander pressure chamber.

3.2

L213/214 see my concern above.

Table 1 Did you notice the differences in hydrogen isotopes for beech SPC and CVDTwB and compare that to the oxygen isotopes? There is no difference in delta18O for CVDTwB and SPC (-5.75 SPC and -5.74 CVDTwB) but 8permil for the same comparison of delta2H. That might just be the bias for your cryogenic extraction line. It is a finding worth mentioning in the discussion when elaborating on the cryo bias.

Results:

L298 – 301 these sentences say the same thing twice, please rephrase

Discussion

I like the discussion now. However, please consider my general comment for section 5.2

Conluding remarks

As I already said in my former reviews, I think this reads like another summary and not a conclusion. I would suggest deleting it.
Also, Line504 "SPC is not an alternative to CVD…" directly contradicts the following lines suggesting using SPC when interested in accessing transpiration. It sounds like the better alternative to me.

---

## Author Response (AR2)

**Response to the editor's and the reviewers' comments**

Dear authors,

Thanks again for submitting your revised manuscript. Two reviewers (one new) have provided detailed comments. They recognise that you made improvements to your previous version but both still have major concerns, in particular regarding the methodology and discussion of your work. I'd like to ask you to consider these reviews. With regards to the methodology, there are a few clarifications needed and please also take care of the comments around the weighted lc-excess and the measurement uncertainty around the lc-excess. Your discussion could be strengthened in particular on the potential reasons for the differences found between methods and species, also considering the work by Barbeta et al 2021, Chen et al 2020 and Allen and Kirchner (2021). I look forward to receiving your revised manuscript.

Best wishes,
Josie Geris

We would like to thank the editor and the reviewers for the time they spent on our manuscript and for the valuable suggestions. We report the reviewer's comments in black and their response below in blue. Lines (L) refer to the revised manuscript with no track changes.

**Response to reviewer 1**
Review Hess Zuecco et al.
Much improved version of the manuscript, thanks.

**Abstract.** No comments

**Introduction:**
I think you greatly improved the introduction and I have only some minor comments:
Line 39-40: water representative of transpiration needed in ecohydrological studies. With your findings the method of choice would be SPC. Pick that thread up in the discussion again.
We thank the reviewer for this suggestion, that we implemented at L404.

Line 80-81: Barbeta et al. observed this depletion, but the problem that it was related to CVD and not the plant fractionation was identified by Chen et al. 2020, and Allen and Kirchner 2021). If you could add these references (see below) and put them in context with Barbeta et al. 2019 that would be great.
We thank the reviewer for this suggestion, that we implemented at L87-96. Note that Allen and Kirchner (2021) HESS now became Allen and Kirchner (2022) HYP.

**Site descriptions:**
Line 109 and 113 what do you want to tell the reader with the Engel et al. in review reference? To look for a better site description in there? I think it is irrelevant and I would delete it, especially because this has not been published yet.

As suggested by the reviewer, we removed the reference since Engel et al. has not been published yet.

Line 125 again, why is it important to have this reference here? I think your site description is sufficient as is, but if you insist on adding a reference here, please add what it is for.
As suggested by the reviewer, we removed the reference at this line.

Table 1 belongs in the Results section, but while looking at it, how do you deal with the tiny deviations from the LMWL (i.e. the small lc values)? See (Landwehr and Coplen 2006) who indicate that lc's smaller than can be detected by the measurement precision are considered not to differ from LMWL (here , it looks like this could be the case for Alder CVDWC, Chestnut CVD-T, Beech SPC and CVD-TwB and -TcT and maybe even more) could you add the calculations for S in regard to the measurement precision of the used instruments (as you write in the MM 2.5permil for delta2H and 0.1permil for delta18O) and then indicate in parentheses which values are and are not different from the LMWL in your case? It would give the reader the chance to directly assess differences.
We thank the reviewer for the useful suggestion. We computed S and modified Table 1 accordingly, we have now moved Table 1 and updated the Results section (we did not add parentheses in Table 1).

Line 136 and again, emphasize what you want to say with adding these references here, or delete them if they are not giving additional information.
In the revised manuscript, we kept only the key references to the experimental catchment.

**Material and Methods:**
Line 234 the landwehr and Coplen reference is dated to 2004 in your manuscript, however when I checked it again on google scholar and web of science the date is 2006, please change it to the correct date.
We thank the reviewer for finding this mistake, that we have corrected.

Generally, I am missing the explanation of how and why you weighted the means of the measurements? I know it says it in the header of table 1 but it should be part of the MM for completeness sake. What did you define as the mean extracted volume? Did that then differ for each method or each species or each tissue? And if so, what is the benefit of weighting them, the measurement device takes the same amount of sample for injection, right?
The comment of the reviewer on the first round of reviews was not very clear, and we misunderstood his/her suggestion. In the revised manuscript, we decided to report the median isotopic composition, as it was in the original submission. However, we kept the information about the extracted volume because it gives an idea about the plant water amount that were extracted from the various sample types.

Also, I think I was misunderstood in the last revision round when I asked for a volume weighted isotopic composition, I meant if you could do a mixing ratio calculation where (in theory) you would mix the samples CVD_L and CVD_TWB together (weighted by each their volume) and get a mixed isotope signal that would in theory then represent the same tissue as when you extracted one twig

with bark and leaves using the SPC. But since you do not see any co-extraction of water from the leaves in the SPC samples, I can agree how you would most closely compare them to CVD-TwB. However, that does not mean they are the same tissue type.

We are sorry we misunderstood the reviewer's indications on the previous version of the manuscript, we re-read it now at the light of this further comment, and we understand what the reviewer meant. We agree with this new comment by the reviewer that even if the mixing ratio calculation between CVD-L and CVD-TwB returned a mixed isotopic signature close to that extracted from leaves and twigs using the SPC method, we would not have conclusive indications about the possible same tissue type. Therefore, we prefer avoiding speculative analysis and discussion, and did not perform the mixing ratio calculation.

**Results:**

Figure 4: If possible, it would be great if you could change the pink line and text to e.g. grey but that's just a personal preference, I feel the pink is too much, maybe other readers think so too? Also, it would be reader friendly if you could plot the leaf data as an inset or a separate column and therewith zoom in to the data plotting closer to the LMWL. Especially looking at the Laas data, I think the figure would benefit greatly from a higher resolution on the potentially non-enriched data.

We thank the reviewer for this suggestion, that we implemented in the revised manuscript.

Line 297: I think once you correct for the measurement precision this will not be distinctly different from 0, so I think it is important that you add this S (as standard deviation) according to Landwehr and Coplen 2006.

We thank the reviewer for the comment, and we think it is a good suggestion. We have revised the text based on the new results (please see L310-317).

**Discussion:**

Figure 8 should go to the last page of the result section.

Yes, we agree with the reviewer, but this issue depend on the LaTeX compiler and the template used for the journal. We will check the position of all figures again before the publication of the manuscript.

I would switch sections 5.1 and 5.2 as it is logical to first read your assessment about the differences and therewith the answer to your main question: do these methods yield different results, and then move on with the method of choice discussion in relevance to ecohydrological questions.

Thank you for this comment. This suggestion makes sense and we considered switching the two sections. However, we think that section 5.1 includes general information, whereas section 5.2 goes in more details addressing the differences between the two techniques and, importantly, the possible implications for isotope-based ecohydrological studies. In order to better stress this focus, we renamed the title of section 5.2 as follows: "Difference between the two techniques and implications for ecohydrological studies".

Line 333 -336: This could also be related to the contamination by phloem, as you did not peel it from the twig. Please add this information here.

Thank you for the comment. We added the information at L354-355.

Line 347: again, I think if you would "correct" the lc-excess values by the precision of your measurement device, these values will be indistinguishable from 0 and therewith indistinguishable from the LMWL.

We thank the reviewer for the comment. Please see the revised text at L364-367.

Lines 351-352: please discuss this also in relation to (Chen et al. 2020, Allen and Kirchner 2021)

Thank you for suggesting these recent works, we are aware of them. The paper by Chen et al. (2020) was heavily criticized due to some possible lack of statistical robustness. However, their results are important as suggest cautiousness when interpreting CVD-extracted stem values due to possible artifacts implicit in the method. We believe that this reference does not fit this part of the discussion as here we reported more depleted isotopic values for both isotopes compared to SPC, without implying fractionation processes addressed by Chen et al. (2020). Still, we mentioned this study at L90-91 of the revised manuscript as it fits better in that part.

The work by Allen and Kirchner (2021) published on HESSD but not HESS was recently published on Hydrological Processes. We added the reference in the discussion at L399-403.

**Concluding remarks:** no comments

**References:**

Allen S and Kirchner J 2021. Potential effects of cryogenic extraction biases on inferences drawn from xylem water deuterium isotope ratios: case studies using stable isotopes to infer plant water sources. Hydrol. Earth Syst. Sci. Discuss.: 1–15.

Chen Y, Helliker BR, Tang X, Li F, Zhou Y, and Song X 2020. Stem water cryogenic extraction biases estimation in deuterium isotope composition of plant source water. Proc. Natl. Acad. Sci. 117: 33345–33350.

Landwehr JM and Coplen TB 2006. Line-conditioned excess: a new method for characterizing stable hydrogen and oxygen isotope ratios in hydrologic systems. Int. Conf. Isot. Environ. Stud.: 132–135.
* * *
**Response to reviewer 2**

This paper addresses very important issue in ecohydrology and particularly for water isotope-based plant water source studies. The authors compare two methods to extract plant water: Scholander-type pressure chamber water extraction (SPC) and the most commonly used method, cryogenic vacuum distillation (CVD). I believe this study is much needed, as the community is looking for a standardized protocol in plant water source studies methods to potentially solve the generalized mismatch between soil and plant water found in many studies. However, I have major concerns about the methodology used and the current version of the paper that should be addressed before publication.

Firstly, from my point of view the reasons explaining why SPC and CVD show different isotopic composition are not profoundly discussed. Particularly, I miss two very important and current references addressing this issue that should be taken into consideration and discussed doubtlessly: (1) Barbeta et al. (2021) New Phyt, uses a novel method (special centrifuge) to separate vessel water from other water in the stem and compare these two differentiate water pools with 'bulk' water extracted by CVD; (2) Chen et al. (2020) PNAS compared CVD extractions to water vapour

transpiration measurements (with several comments associated in PNAS and Hess). Both studies show a more enriched value of the xylem and transpired water compared to CVD (like Zuecco et al.) but associate this offset to different reasons (1) heterogeneity of water pools/domains inside the stem (2) the H exchange between organic compounds (mostly cellulose) and water during CVD water extraction. I encourage the authors to take into consideration the discussion of these papers and I believe this can be of much help to solve some of the concerns that the previous reviewers and myself have.

We thank the reviewer for the comment and suggestion. In the revised manuscript, we discussed the findings by Barbeta et al. (2021) and Chen et al. (2020) at L87-94, L373-375 and L399-403.

Additionally, I have some concerns about the methodology used:

• I don't understand the reasons behind sampling at sunset. If the authors wanted to have the maximum hydration of the tissues (in order to get enough water for analysis) they should have conducted the sampling campaign at predawn. If they wanted to sample sap water that was transpired they should have conducted the sampling campaign at the moment of maximum transpiration (mid-morning if the conditions where very dry). I cannot understand of what conditions would be representative the sunset sampling. Maybe it was just because logistic reasons but this should be either well discussed or considered a limitation of the study. Particularly, I think about the many studies, p. ex. Pfaust et al. (2015) Tree Phys, showing daily dehydration-rehydration cycles inside the stems (with a key function of parenchyma rays in this paper). In this regard, Barbeta et al. 2021 showed also an exchange between xylem and other 'tissue' water in the stems. With this comment I mean that xylem water collected by SPC could be influenced by other water domains in the stem depending on the time of the day.

We thank the reviewer for his/her comment. Sampling at the sunset was carried out for logistic reasons, but we agree that it could represent a limitation of the study. So, we added new sentences at L439-446.

• In the same line as the other referees, I wonder about the contamination from phloem or other living cells during SPC sap extraction, particularly when the authors say that the extracted water was coloured (to me an indicator that there is mixing of other water in the stem apart from sap). I think no more analysis can be conducted in the samples from this study but if the authors want to promote this method this is an important issue to solve. Maybe it would be necessary not to reach too high pressures (so, the method would not be valid in very dry conditions) and analyse for organic compounds as a proxy of other tissues contamination. In this regard, in both Geißler et al. (2019) and Barbeta et al. (2021) xylem water was always less contaminated by organics than bulk (CVD). I would also suggest to the authors to make a test conducting SPC with and without leaves, bark and phloem and check differences in water isotopes and the presence of organic compounds.

We thank the reviewer for his/her comment. We already mentioned in section 5.3 that the contamination from phloem or other living cells represents a potential limitation of our experimental setup, and furthermore, the quantification of organic compounds should be performed in future studies.

• The sampling of CVD-WC or CVD-TwB, as now addressed, does not give any relevant information to the study. I would suggest to remove these results or justify better their significance.

We thank the reviewer for this comment. Our main goal was to compare the two techniques for water extractions from different plant tissues and, as such, we think it is useful to present a wide range of plant tissues not to leave any curiosity in the reader. Therefore, we kept the wood cores and twigs with bark samples in the paper.

• Do you have any soil data from these or previous campaigns? It would be good to see if soil water fitted better to SPC or CVD samples. Anyway, the overlap of SPC values to the LMWL (lower lc-excess) would indicate to me that the isotopic values for SPC are closer to the water that plants uptake and transpire than the CVD ones.

In principle, we agree with the reviewer that it would be interesting to see how the different water samples extracted through the two techniques compare to soil water samples. However, the degree of similarity and overlapping between xylem and soil water isotopic composition does not always give clear indications about the sources of plant transpiration for a variety of reasons, including possible fractionation of xylem samples and/or shallow soil water samples; presence of other sources for tree transpiration in addition to soil water; uncertainty in collecting representative soil water samples (see Beyer and Penna, 2021). We do have soil water samples for some of the study sites but, for these reasons, we believe that including them into the analysis would increase the complexity of the interpretation of the comparison experiment between two techniques and, ultimately, make the main message of the paper more cloudy.

• Extraction times: Did you take note of the extraction times for each sample? It would be good to check if there is a relation between extraction time and evaporation. However, this relationship, could be also associated to a more enriched sap water inside the vessels (Martín-Gómez et al. 2016. Tree Phys)

Unfortunately, we do not have data about all extraction times, and therefore, we cannot add such analysis in our manuscript.

• I agree that the fact that the isotope composition of SPC samples is not enriched like the CVD leaves, suggest that during the SPC water extraction no strong contamination from leaves water was happening. However, I would suggest the authors an additional use for the CVD-leaves isotopic composition. By building a regression line of the CVD-leaves it's possible to retrieve the origin of the leaf evaporative water line; and check the possible water sources of the leaves (Barbeta et al. 2021).

We think the reviewer refers to the recent work by Benettin et al. (2021), and not by Barbeta et al. (2021). Benettin et al. (2021) used a backward evaporation model to map leaf water back to its individual precipitation event water sources. This is an extremely interesting approach that can open new research ways in isotope ecohydrology. However, Benettin et al. (2021) provided only a preliminary proof of concept and this method deserves to be carefully tested with specifically planned experimental designs that, we believe, are not offered by our current dataset. Some authors included in this research are currently working with Benettin to test this approach under specific conditions. Furthermore, we have too few leaf water samples, particularly for Ahr/Aurino and Ressi, to apply a meaningful linear regression analysis.

• There is a lack of analysis and discussion about species-specific differences between the methods. For example, for beech the difference between SPC and CVD is smaller than in the other species.

Could these differences be associated to wood anatomy (vessel area, wood density, parenchyma volume fraction…) or wood properties (quantity of lignin…)? Please check numerically and discuss it.

We agree with the reviewer that it would be interesting and of practical importance to analyze whether the differences in the isotope values returned by the two extraction techniques might be related to species characteristics, such as wood anatomy or wood property. However, this would require an array of wood data that we do not have. Relying on information and data reported in the literature would not be appropriate to address these aspects in a robust way and speculative discussion would follow. We prefer avoiding this.

Apart from these general comments, that will necessarily change the content and structure of the paper in several parts (please include them along all the document), I have some minor comments outlined in the text:

L18. CVD-leaves is not comparable to SPC in twigs.

Thank you for the comment. We removed the parentheses.

L21: soil water sources

We added "soil".

L21: rephrase xylem water transpiring during the sampling day. Maybe just 'sap water' or 'xylem water'

Done.

L36-37. For soils I would not say 'a little'. Update the references for plant water extraction techniques

Thank you for the comment. We rephrased the sentence.

L45. Update the techniques (Barbeta et al. 2021, Zhao et al. 2016)

Done.

L48. Also, water within cell walls.

Done.

L48 and L52. Consider that water inside living cells might not be just different because of the age of the water but also because of fractionation inside the plant (aquaporins)

Thank you for the comment. We integrated the sentence at L51.

L86. Which other ecophysiological method could be used for this purpose?

Thank you for the comment. However, the aim of our study was to compare SPC and the widely adopted CVD and, therefore, we focus on these two methods in the introduction.

L92. Any of these studies checked if the isotopic composition of SPC extracted water fitted with soil water?

Yes, the mentioned studies considered the isotopic composition of both soil water and plant water extracted by SPC. However, we think this detail is not of great important for our introduction.

L99. Urgently needed to find the best method to sample transpiration water.
Thank you for the suggestion, that we have implemented in the text.

L101. Consider again differences not only associated to different ages but also to internal cell fractionation. In Barbeta et al. 2021, they prove that the stem water (both xylem and other tissue water) is replaced within 3 days of well water conditions. Besides, they observe a more or less constant offset between sap and bulk water.
Thank you for the suggestion. Please see the revised text at L113.

Table 1 and analysis. Are there species-specific differences in the quantity of water collected in the SPC and CVD twig samples?
Thank you for the comment. We have added the results of the test at L278-281.

Table 1. Did you take note of the pressure and time of the extraction? If you did, I would include the values in the table.
Unfortunately, we do not have notes for pressure and extraction times of each sample, and therefore, we cannot report these details in the table.

L150. Rephrase 'bad injections', change for "problems with the analyser"?
Thank you for the comment. We rephrased with "injection issue".

L162. Please explain "SPC contains a cutting board..." consider removing it or putting the sentence in context
We removed this part of the sentence.

L162. Consider rephrasing "The set up consisted..." one or more leaves alone? I guess you meant "one branch/twig with one or more leaves..."
Thank you for the comment. We rephrased the sentence.

L172. It would be good to have the water deficit conditions of every site (i.e. water potential of every species). Table 1?
We agree with the reviewer that this an interesting detail. Unfortunately, we did not take notes of water deficit for all samples.

L174. Did you consider not to flow all the water out of the twig samples with SPC? Did you do this for any physiological reason or because you were not collecting enough water?
We collected all the water flowing out from the twigs because the volumes were not enough for the isotopic analyses.

L175. Wood anatomy, particularly xylem anatomy could explain also the different extraction times? For example, in Ressi, did you find differences between the two species?

We agree with the reviewer that it would be interesting to analyze whether the differences in the isotope values returned by the two extraction techniques might be related to species characteristics, such as wood anatomy. However, this would require an array of wood data that we do not have. Relying on information and data reported in the literature would not be appropriate to address these aspects in a robust way and speculative discussion would follow. Therefore, we prefer avoiding this.

L182. What does it mean fractionated in this sentence? Please rephrase or explain
We removed "fractionated".

L185-186. Rephrase this part also. I don't understand the 'use' of CVD total plant water
We removed "total".

L192-195. I would put this paragraph in the introduction or discussion, not in methods
We moved the sentence to L389-393.

L238. Evaporation + other fractionation processes
We added "other fractionation processes".

Figure 4. Leaf evaporative water line. The legend for the colours is lacking
Unfortunately, we have too few samples to apply the simple linear regression, or the method proposed by Benettin et al. (2021). Furthermore, please see our previous reply to the specific comment on the leaf water evaporation line.
Thank you for noticing that the legend for the colours was missing; we fixed this adding a sentence in the caption of the figure.

Figure 5 and 6 are redundant to lc-excess in Figure 7. Consider to remove or move them to Supplementary Material
We disagree with the reviewer because Figures 5 and 6 are important to determine which CVD samples are more similar to the SPC samples. So, we decided to keep them in the revised manuscript.

L293. Again, I don't think you can compare CVD-leaves with SPC.
Here we do not compare CVD-L with SPC samples. Furthermore, one of the previous reviewers asked to add more comments on the CVD-L data presented in the manuscript.

Discussion: you write the study limitations in two different sections. I would suggest to reorganise all the discussion and refocus your main message with the suggestions I wrote in the first part of the review.
We think that the three sections of the manuscript present different arguments. Furthermore, we heavily revised the discussion following the suggestions of the two previous reviewers, and based on their comments we added a specific section on the limitations of our study. As suggested by the reviewer, we discussed the findings of Barbeta et al. (2021) and Chen et al. (2020) in the newly revised manuscript.

L342-343. Explain better that statement or remove it. I think, even though CVD from twigs, trunk and branches are closer to LMWL, this is not unequivocal prove of no fractionation. If you look into the results in detail you can see that the offset for d2H is generally larger than for d18O, and not drawing an evaporative line, which could indicate a fractionation caused for a different process than evaporation (mixing, biochemical processes, etc.). Obviously, the leaves are the most fractionated/evaporated samples. Please clarify the term fractionation along the document.

We thank you the reviewer for this comment, we agree with her/him that samples plotting on or close to the LMWL do not necessary imply that no fractionation occurred. Throughout the manuscript we carefully checked if the term "fractionation" was used appropriately. Here, we rephrased as follows: "Our results showed that twig samples (with leaves) obtained by SPC, differently from the CVD-L samples, did not show any offset compared to the LMWL of each site (Fig. 4)".

L348. Please rephrase. With this sentence I could understand that all the methods could be valid. Again, clarify fractionation.

Please see the rephrased sentence at L367-370.

L369. Remove the sentence into () or the word "particularly"

Done.

**References:**

Barbeta, A., Burlett, R., Martín-Gómez, P., Fréjaville, B., Devert, N., Wingate, L., Domec, J.-C. and Ogée, J. (2021), Evidence for distinct isotopic compositions of sap and tissue water in tree stems: consequences for plant water source identification. New Phytol. https://doi.org/10.1111/nph.17857

Chen Y, Helliker BR, Tang X, Li F, Zhou Y, Song X. 2020. Stem water cryogenic extraction biases estimation in deuterium isotope composition of plant source water. Proceedings of the National Academy of Sciences, USA 117: 33345– 33350.

Martín-Gómez P, Serrano L, Ferrio JP (2017) Short-term dynamics of evaporative enrichment of xylem water in woody stems: implications for ecohydrology. Tree Physiology, 37, 511-522, http://doi.org/10.1093/treephys/tpw115

Pfautsch S, Renard J, Tjoelker MG, Salih A. 2015. Phloem as capacitor: radial transfer of water into xylem of tree stems occurs via symplastic transport in ray parenchyma. Plant Physiology 167: 963–971.

Zhao L, Wang L, Cernusak LA, Liu X, Xiao H, Zhou M, Zhang S. 2016. Significant difference in hydrogen isotope composition between xylem and tissue water in Populus euphratica. Plant, Cell & Environment 39: 1848– 1857

---

## Author Response (AR3)

**Response to the editor's and the reviewers' comments**

Dear Authors, could you please consider the minor comments from Reviewer 2 and also specifically the comment regarding like-for-like comparisons of the methods please?

We would like to thank the editor and the two reviewers for the time they spent on our manuscript and for the valuable suggestions. We welcomed the comments by reviewer 2 and the editor suggesting to explain that the SPC vs. CVD-TwB comparison is the one which makes more sense and is used as kind of reference. At the same time, we decided to keep the comparison between all other water types because we believe this make the paper more impacting and more useful for the ecohydrological community.

We report the reviewer's comments in black and our responses below in blue.

**Response to reviewer 2**

Review 3 Hess-2020-446

A comparative study of plant water extraction methods for isotopic analyses: Scholander-type pressure chamber vs. cryogenic vacuum distillation

**General comments:**

I will dive right in, since I summarized the manuscript the last two times, I reviewed it. I think I have to apologize for not making my points clear enough, I will try to here:

In the results section of this revised version of the manuscript the authors compare every CVD extracted tissue to SPC extracted tissue. The only comparison I think the authors can argue to be valid is that of SPC and CVDTwB and therefore, in my opinion, the only comparison that should be carried out and communicated in the results section. It is an **a priori** assumption that the methods are **not** comparable for all the other plant tissues you extracted with CVD and therefore it is not an issue to be discussed but a comparison that should not be carried out in the first place. Therefore my suggestion would be for you to take this under consideration and revise the manuscript accordingly. Especially in these sections 3.4 Data analysis, 4 Results, and 5. Discussion you should refrain from comparing all but CVDTwB and SPC. I'm very sorry if this is my fault for not communicating this strongly enough. I think you can very well communicate the other CVD extracted data as a comparison of plant tissue differences but not to compare the two methods.

We would like to thank the reviewer for this further round of revision of our manuscript. We agree with the reviewer that we had a priori assumptions about all possible comparisons. Therefore, as suggested, in the revised manuscript we focused more on the comparison between SPC and CVD-TwB, and we clarified that this represents the "reference comparison" in our manuscript. However, we prefer to keep also the comparison regarding plant water extracted by different tissues using CVD. Our proposal is based on the fact that it would be quite useful for the scientific community to know whether there are significant differences in the isotopic composition of such extracted waters, and our manuscript helps shed light into this topic.

I would also again encourage you to have a native speaker check the revised version.

We have a native speaker in the authorship, and we performed a careful polishing of the English language.

**Specific comments:**

**Introduction:**

L48 the extraction of water directly from the stem due to positive pressure on the inside is only possible for very few tree species, otherwise no one would have to ever extract any plant tissues. Please make sure to clarify this when adding that reference (Zhao 2016)

Done.

L64 which two versions of CVD are you referring to here?

We refer to both systems used in Millar et al. (2018). We think that there is no need to specify the characteristics of the two systems here.

L94 check formatting of references

The formatting of the references is correct.

L114ff Here you switch to present tense while the rest of the introduction is written in past tense. I would suggest choosing either one consistently throughout the manuscript.

We changed the tense of these sentences as the reviewer requested. However, we think that for general (and generalizable) statements the present tense is more adequate, as thus we do not use the past tense only throughout the manuscript.

**Material and Methods**

L188 The bark and the leaves remain attached to the twig on what basis? In physiology the bark is usually removed to avoid phloem contamination. If you insist on keeping this reference here, I would ask you to dig a little bit into physiology research literature and at least mention that this is not the standard procedure when extracting xylem using the Scholander pressure chamber.

We agree with this comment. As requested by the reviewer, in the revised manuscript we added a sentence about the fact that this is not the standard procedure for the extraction of xylem water using the Scholander pressure chamber.

3.2

L213/214 see my concern above.

We revised the text according to the general comment.

Table 1 Did you notice the differences in hydrogen isotopes for beech SPC and CVDTwB and compare that to the oxygen isotopes? There is no difference in delta18O for CVDTwB and SPC (-5.75 SPC and -5.74 CVDTwB) but 8permil for the same comparison of delta2H. That might just be the bias for your cryogenic extraction line. It is a finding worth mentioning in the discussion when elaborating on the cryo bias.

We thank the reviewer for this suggestion, that we implemented in the discussion.

**Results:**

L298 – 301 these sentences say the same thing twice, please rephrase

The two sentences convey the results of two statistical tests; in the first sentence we reported the result of the Friedman repeated measures analysis of variance on ranks, whereas in the second sentence we specified the results of the Tukey test. Therefore, we prefer to keep the text as it is.

**Discussion**
I like the discussion now. However, please consider my general comment for section 5.2
We are glad you appreciated the revised discussion. We made few more changes according to the general comments.

**Concluding remarks**
As I already said in my former reviews, I think this reads like another summary and not a conclusion. I would suggest deleting it.
In the first revision, we tried to reduce this section and to consider all comments of the reviewers. We think this section does not need a further thorough revision (although we removed the first sentences of this section), except for the following minor comment.

Also, Line504 "SPC is not an alternative to CVD…" directly contradicts the following lines suggesting using SPC when interested in accessing transpiration. It sounds like the better alternative to me.
We would like to thank the reviewer for pointing out this unclear phrasing that we modified.
* * *
**Response to reviewer 3**
I think the authors greatly improved the manuscript. All the 'weaker' points have been clarified and the document reads very well. I believe it will be an important paper for the community.
I do not have anything else to report. Thank you for considering my comments.
We would like to thank the reviewer for his/her comments and suggestions, and for appreciating the revised manuscript.